# Hydrogen Cryomagnetic a Common Solution for Metallic and Oxide Superconductors

**DOI:** 10.3390/ma18153665

**Published:** 2025-08-04

**Authors:** Bartlomiej Andrzej Glowacki

**Affiliations:** 1Institute of Power Engineering, National Research Institute, DZE-2, ul. Mory 8, 01-330 Warsaw, Poland; bag10@cam.ac.uk or bartlomiej.glowacki@ien.com.pl; 2Department of Materials Science and Metallurgy, University of Cambridge, 27 Charles Babbage Road, Cambridge CB3 0FS, UK

**Keywords:** liquid hydrogen, metallic LTS superconductors, ceramic HTS superconductors

## Abstract

This article examines the physical properties, performance metrics, and cooling requirements of a range of superconducting materials, with a particular focus on their compatibility with hydrogen-based cryogenic systems. It analyses recent developments and challenges in this field, and considers how hydrogen cryomagnetic could transform superconducting technologies, making them economically viable and environmentally sustainable for a variety of critical applications. The discussion aims to provide insights into the intersection of metallic and ceramic superconductors with the hydrogen economy and to chart a path towards scalable and impactful solutions in the energy sector.

## 1. Introduction

For over a century, the study of superconducting materials has been a dominant area of scientific enquiry and engineering innovation, offering significant potential for the advancement of power engineering and the development of advanced technologies. The development of metallic low temperature superconductors (LTS) and oxide high temperature superconductors (HTS) has facilitated the realisation of several practical applications, including traction transformers [1], power cables [2], fault current limiters (FCLs) [3], magnetic ingot heaters [4], magnetic separators [5], superconducting energy storage devices (SMES) [3,6], motors [7], and Tokamak fusion reactors [8] and many others [9,10]. Even though superconducting materials are now close to being ready for large-scale deployment, the high cost and complexity of cooling them to their operational low temperatures remain significant barriers to their widespread adoption.

The thermophysical properties of liquid hydrogen listed in Table 1, particularly its low boiling point, high specific energy, and substantial latent heat, make it a uniquely capable medium among other cryogenic liquids such as helium and nitrogen for combined energy storage and cryogenic applications when leveraged in superconducting systems.

LH_2_ can reduce cooling complexity, lower energy consumption in multistage refrigeration, and enable the integration of fuel and thermal management into a single infrastructure. As decarbonized energy systems advance, these multifunctional properties position LH_2_ as both a key energy carrier and an enabling refrigerant for emerging superconducting technologies.

In the context of the current global scenario, in which the challenges of climate change and energy crises demand urgent and sustainable solutions, the integration of the hydrogen economy presents a promising pathway. The utilization of low-cost liquid hydrogen (LH_2_) as a coolant represents a viable and complementary approach to addressing these challenges, which has been termed ‘hydrogen cryomagnetic’. The utilization of LH_2_ as a coolant enables the efficient and cost-effective cooling of metallic and oxide superconductors, thereby unlocking their full potential for a variety of applications. Hydrogen cryomagnetic technology represents an innovative convergence of two advanced scientific domains: superconductivity and cryogenic energy hydrogen systems. This technology utilizes both metallic and ceramic superconductors, materials that can perform well in liquid hydrogen as a cryogenic medium. Liquid hydrogen plays a dual role: it serves not only as a potential clean energy carrier but also as a coolant that maintains superconductors at their required low operating temperatures. By leveraging hydrogen’s extremely low boiling point (14–32 K), this approach eliminates the need for more expensive or complex cryogenic systems, such as those based on liquid helium; however, at the initial stage of indirect cooling, i-LH_2_, with a pressurised helium loop, may be the more obvious option. The integration of superconductors with improved in field performance by maximisation of the pinning at higher magnetic flux densities and liquid hydrogen, opens pathways to create ultra-efficient magnetic systems, which may include cryomagnets for energy storage (SMES—superconducting magnetic energy storage), magnetic levitation for transport, high-field magnets for research and medical imaging, and plasma confinement in fusion energy devices.

## 2. Hydrogen Cryomagnetic

The primary motivation for cryogenic cooling of magnets—whether direct current (DC), alternatig current (AC), pulsed, superconducting, or resistive—is to enhance the electrical conductivity of the conductor in the most cost-effective manner. Traditionally, liquid helium (LHe) or compressed helium gas has been employed for such cooling applications. Monte Carlo simulations conducted in 1990 (Figure 1) indicated a general trend of increasing helium wholesale prices over time [11]. Price fluctuations are primarily driven by two factors: production costs and the imbalance between supply and demand, the latter referred to in the model as “excess demand”. Helium production costs are significantly influenced, though not exclusively determined, by energy prices (e.g., electricity), helium concentration (particularly in non-cryogenic separation processes), labour, raw material costs, and royalties [12].

As discussed in the model validation section, the initial settings for key parameters were based solely on quantitative data, excluding price behaviour (Figure 1). This decision was made because while quantitative (supply/demand) data are relatively robust and consistent globally, pricing is subject to greater uncertainty and data inconsistencies. Consequently, some mismatch between historical and simulated price data in Figure 1 is expected. However, what is more concerning is the observed divergence in both the phenomenological behaviour and magnitude between real and simulated data.

System dynamics and Monte Carlo simulations were successfully employed to predict helium price trends, showing excellent agreement with historical market data over nearly two decades. This research, based on datasets from British Oxygen Company (BOC) and the Culham Fusion Centre, demonstrated that until 2009, the simulated helium prices closely tracked real-world values. However, after 2009, actual helium prices began to diverge sharply from the predicted trajectory, rising far more steeply than anticipated. As of today, helium prices have escalated to such an extent that simulation-based forecasting is no longer meaningful; current prices are so disproportionately high that even helium extraction from ambient air is being reconsidered as a viable option. This deviation is driven by multiple factors: the rise of shale gas production, which contains negligible helium and thus excludes helium recovery from consideration, and the overwhelming global demand for natural gas. As a result, helium recovery during LNG production has become a secondary priority, further tightening global helium supply.

For large-scale installations, the forecasted helium shortage and the already observed price increase by more than an order of magnitude necessitate the development of new cooling policy frameworks [11,12]. Figure 1 indicates that helium prices are approaching USD ~1000 per Mcf, nearing the theoretical cost of extracting helium from ambient air (see crossover point in Figure 1). However, despite the elevated market price, helium extraction from air remains commercially unviable. This is primarily because current air separation units are optimized for oxygen and nitrogen, not helium.

The cost of helium extraction from air (where helium is present at ~5.2 ppm) depends on several factors, including the extraction technology, desired purity levels, energy input, labour, and transportation. The process is technologically complex and rapidly evolving, making accurate cost estimation highly sensitive to these parameters. In applications such as cooling low-temperature superconductors (LTSs) for quantum electronics in the millikelvin (mK) range, or in medical imaging systems like MRI that use metallic superconductors, the high cost of cryogenic cooling must be considered. The required input power from room-temperature machinery, expressed in kilowatts (kW), to generate just 1 Watt (W) of cooling at cryogenic temperatures is substantial. Figure 2 illustrates the minimum input power needed at room temperature to achieve 1 W of cryogenic cooling across various temperature ranges, alongside the associated estimated equipment costs and a brief description of available cooling technologies.

### Hydrogen Cryomagnetic Cooling: Indirect and Direct LH_2_ Approaches

It is evident from Figure 2 that at liquid hydrogen (LH_2_) temperatures, the operation of cryogenic cooling equipment becomes economically viable. In contrast to the shrinking and increasingly expensive liquid helium (LHe) market (see Figure 1), the hydrogen economy is rapidly expanding, as projected in Figure 3. With growing global investment and infrastructure, LH_2_ is expected to become widely available. This enables the broader adoption of indirect LH_2_ cooling (i-LH_2_) as a standard solution for both metallic and oxide superconductors under the emerging domain of hydrogen cryomagnetic. Provided that hydrogen safety technologies continue to mature, direct LH_2_ cooling (d-LH_2_) may also become feasible. This would allow for hybrid cryogenic systems in which LH_2_ serves dually as a coolant and energy vector. To evaluate these developments, we first examine metallic superconductors and argue that high-temperature superconductors (HTSs) should prioritize i-LH_2_ as a cost-effective alternative, based on cooling cost data presented in Figure 2, to fully exploit HTS’s potential. To date, the liquid hydrogen option has received limited attention, primarily due to the challenges associated with direct LH_2_ cooling. However, an established and practical intermediate approach is to use a closed-cycle, low-loss loop of compressed helium gas. This system can provide effective cooling at 20 K (or potentially down to 15 K), allowing users to benefit from LH_2_’s thermal performance without the direct safety risks associated with hydrogen handling [13].

Unlike boiling cryogens, a compressed helium gas loop avoids issues like critical heat flux. However, it may exhibit lower local heat transfer coefficients. These can be improved significantly by engineering micro-channels into the system substrates, which could offer superior thermal coupling, potentially outperforming conventional LHe-based cooling systems. A comparison of technology implications between helium-cooled indirect hydrogen systems (i-LH_2_) and conventional helium or nitrogen-based cooling is presented in [3], where research needs and opportunities are also outlined.

As reported by McDonald et al. [14], the power required to operate a copper magnet can be dramatically reduced by lowering the operating temperature. However, below approximately 30 K, copper’s electrical resistance decreases only marginally, establishing this temperature as the economic threshold for high-field, high-current resistive magnets. At 1 bar pressure, liquid hydrogen provides a natural coolant for these systems. Yet, due to safety concerns, direct cooling of copper conductors with LH_2_ is generally inadvisable.

An effective alternative is to cool the magnet with helium gas at ~30 K, itself chilled via an external heat exchanger using LH_2_, an indirect LH_2_ (i-LH_2_) scheme. A design based on this concept was developed for the 15 T pulsed solenoid used in the MERIT experiment at CERN (Figure 4). In this context, McDonald et al. [14] conducted economic analyses of cooling efficiency for large, resistive electromagnets, comparing systems based on LHe, LH_2_, and LN_2_. Their findings reinforce the feasibility of LH_2_-cooled systems for large-scale high-power magnetic devices.

Taking into account the prices in 2002—USD 0.53 per liter for liquid hydrogen (LH_2_) and USD 3.00 per liter for liquid helium (LHe)—it is evident that cooling copper magnets with LHe was approximately 70 times more expensive than indirect cooling using i-LH_2_. Over the following decades, the price of LHe increased by more than an order of magnitude (see Figure 1), while the price of LH_2_ declined. By 2023, the cost of helium-based cooling exceeded that of i-LH_2_ by a factor of 1700, and today this differential has grown to more than 2000 times. Importantly, among potential cryogenic fluids: helium, nitrogen, and hydrogen, only hydrogen functions as both a cryogen and an energy carrier. This dual functionality makes LH_2_ an especially attractive choice for cryogenic applications, providing unmatched economic and operational advantages.

There are also practical demonstrations of hydrogen cryomagnetic in the literature. One notable example is presented by Hirabayashi et al. [15], Figure 5, who describe a superconducting magnetic energy storage (SMES) system in which LH_2_ serves two roles: (a) as a cryogen for conduction cooling of the SMES magnet to ~20 K, and (b) as an energy source post-discharge. Following a 60 s discharge event (e.g., after a fault), the same LH_2_ used for magnet cooling can be redirected to power a hydrogen fuel cell. This is illustrated in Figure 5a,b. The superconducting magnet offers rapid response and high input/output electrical power, while LH_2_ ensures high energy density and enables efficient electricity generation through the fuel cell. This synergistic integration of SMES and hydrogen fuel cells offers a compelling path forward for sustainable, high-performance energy systems. Having outlined key examples of i-LH_2_ used in efficient cooling of both normal-conducting and superconducting magnets, we now turn to the potential of direct LH_2_ (d-LH_2_) cooling. In future sections, we will assess the types of superconductors that could feasibly be cooled using d-LH_2_.

Meanwhile, i-LH_2_ systems remain extremely practical. For instance, helium-based cryocoolers can supplement i-LH_2_ systems to achieve operating temperatures in the 4–14 K range. This requires only a modest temperature reduction (e.g., from 20 K to ~10 K) thereby avoiding the need for the expensive and power-intensive technologies previously outlined in Figure 2.

## 3. Superconductors Selection

The diagram in Figure 6 illustrates the historical evolution of superconducting materials, tracing how successive discoveries have progressively increased the critical temperature (T_c_), thereby expanding the scope of economically viable superconducting technologies. Early superconductors, discovered in the early 20th century, were primarily elemental metals and intermetallic alloys with T_c_ values below 10 K. Their operation required cooling with liquid helium (LHe), boiling point 4.2 K, a highly effective but costly and logistically challenging cryogen due to its scarcity and low boiling point. The subsequent discovery of NbTi and Nb_3_Sn in the 1950s and 1960s, with T_c_ values in the 9–18 K range, made practical applications more accessible but still dependent on helium-based systems. A significant paradigm shift occurred in 1986 with the discovery of cuprate high-temperature superconductors (HTSs) such as YBa_2_Cu_3_O_7_ (YBCO), which exhibit T_c_ values exceeding 90 K. These could be cooled using liquid nitrogen (LN_2_, boiling point 77 K), thereby drastically reducing the cost and complexity of cryogenic systems.

More recently, interest has emerged in hydride superconductors operating at T_c_ values above 200 K under high pressures. While promising for fundamental research, their requirement for gigapascal-range pressures limits their practicality for applied superconductivity, especially in energy, magnetic, or transport systems. These materials often display localized superconductivity, rendering them currently unsuitable for large-scale engineering applications.

An often-overlooked intermediate domain lies between these extremes, materials with T_c_ values in the 15–30 K range. These are beyond the reach of LN_2_ cooling but do not require LHe. For such materials, liquid hydrogen (LH_2_, boiling point 20.3 K at 1 bar) presents an attractive and economical alternative. LH_2_ is not only more abundant and significantly less expensive than helium, but it also holds dual value as both a cryogen and an energy carrier, especially appealing in decentralized and hybrid energy systems.

Materials such as magnesium diboride (MgB_2_) and certain engineered Nb_3_(Al,Ge) alloys are highly compatible with LH_2_-based systems. In addition, many HTS materials—such as (RE)Ba_2_Cu_3_O_7_δ (RE = Y, Gd, or mixtures thereof), Ag-(Bi,Pb)_2_Sr_2_Ca_2_Cu_3_O_x_ (Bi-2223), Bi_2_Sr_2_CaCu_2_O_x_ (Bi-2212), and thallium or mercury-based cuprates (e.g., Tl- and Hg-HTS) can achieve optimal J–B–T performance (critical current density vs. magnetic field vs. temperature) at LH_2_ temperatures, compared to degraded performance near LN_2_ temperatures (close to T_c_).

The growing trend in superconductor research and engineering is toward higher T_c_ materials compatible with more economical, higher-temperature cooling methods. However, while LN_2_ is widely available and low-cost, it has no intrinsic energy value as a fuel or chemical energy carrier it cannot release usable energy through combustion or reaction, unlike hydrogen, which is a true energy carrier capable of releasing significant energy when oxidized. The only recoverable energy from LN_2_, analogous to that of LH_2_, comes from its physical expansion during the phase change from liquid to gas. This expansion can be harnessed via a turbine to generate electricity on demand. Moreover, many high-temperature superconducting (HTS) conductors experience significant degradation in their J–B–T performance envelope at the boiling point of LN_2_.

In contrast, LH_2_ occupies a strategic thermal and energetic niche, with excellent latent heat characteristics, making it uniquely suited to support advanced superconducting applications across the 15–30 K range and beyond.

Enhancing superconducting properties, especially increasing T_c_, generally involves increasing material complexity. This includes structural anisotropy, stoichiometric substitution, multistage processing, and superconductive anisotropy (γ), as shown in Figure 7. These complexities, while effective in elevating performance, also impose practical limitations, particularly for AC applications and non-trivial geometries, where simpler materials may still offer advantages [16].

In conclusion, liquid hydrogen emerges as a key enabler of the next generation of superconducting systems. It supports a broad range of materials that are otherwise underserved by traditional cryogens, while simultaneously acting as an energy-dense energy carrier a synergy of thermal and energetic functions that points toward hydrogen cryomagnetic as a cornerstone of future applied superconductivity.

Toxicity is a critical factor that continues to limit the applicability of even the highest-temperature superconductors. Due to stringent EU regulations, mercury-based and thallium-based superconductors, despite their record-high critical temperatures (T_c_) and excellent current density-field-temperature (J-B-T) performance, cannot be manufactured because of their toxicity. Similarly, toxicity concerns are likely to preclude future commercialization of arsenic-based iron pnictides and lead-based Pb-2223 superconductors, as summarized in Table 2.

Consequently, the list of viable applied superconductors is narrowing, currently including only NbTi, Nb_3_Sn, Nb_3_(Al,Ge), MgB_2_, Bi-2212, and REBCO conductors. While NbTi and Nb_3_Sn may benefit from efficient pre-cooling with liquid hydrogen (LH_2_) prior to further cooling with liquid helium (LHe), this article focuses on superconductors whose J-B-T phase diagrams Figure 8 indicate the potential for effective cooling within the 14–33 K range, where LH_2_ becomes a practical and economic cryogen.

We propose that LH_2_ should be broadly recognized as a standard cryogen, particularly suitable for both indirect (i-LH_2_) and direct (d-LH_2_) cooling of superconducting cryomagnetic devices. Indirect cooling via LH_2_–He gas heat exchangers is demonstrated in Figure 4 and Figure 5, while direct immersion cooling offers an efficient alternative for compatible superconducting systems.

Before examining specific metallic superconducting materials in the context of hydrogen cryogenics, it is essential to consider additional techno-economic interdependencies that govern the practical application of superconductors at cryogenic temperatures.

The first consideration is the maximum of J_c_/cost(T), the ratio of critical current density to cost, which defines the economic feasibility of using LH_2_ as a coolant. This relationship is illustrated schematically in Figure 9a, showing an optimal point balancing cooling cost and conductor manufacturing cost.

The second aspect, depicted in Figure 9b, is the critical current density of the superconducting layer only, denoted J_csup_(d, T), where *d* represents the superconducting layer thickness. This property is influenced by layer nucleation, grain growth, and pinning site distribution, all of which can cause significant variation in current density across the thickness, affecting both metallic and ceramic superconductors.

Lastly, Figure 9c presents the engineering critical current density, J_ceng_(d, T), which incorporates the entire conductor structure—superconducting layer, buffer layers (for HTS), substrate, stabilizer, and insulation. Notably, J_ceng_ is generally lower than J_csup_ and reaches its maximum at a larger superconducting layer thickness due to the reduced superconductor-to-conductor cross-sectional ratio. Together, these evaluations support the practical optimization of commercially relevant superconducting wires, including Nb_3_Sn, Nb_3_(Al,Ge), MgB_2_, Bi-2212, and REBCO, for use in advanced cryomagnetic applications.

## 4. Metallic Superconductors

Figure 9 presents a schematic of the critical surface for all practical superconducting materials, defined by three key parameters: critical current density (J_c_), critical temperature (T_c_), and upper critical magnetic flux density (B_c2_). Operating within this critical surface is not merely a guideline; it is an engineering necessity. For any superconducting power application, the boundary function f(J, B, T) fundamentally determines the conditions for safe, stable, and efficient operation.

In superconducting magnets—such as those used in MRI machines or tokamaks—exceeding these critical limits can lead to local transitions to the normal (resistive) state, resulting in magnetic field distortions and potentially catastrophic system failures. Similarly, in superconducting DC or AC power cables cooled by hydrogen at approximately 20 K, surpassing critical limits leads to resistive losses and major grid disruptions. To avoid these risks, engineers incorporate substantial design margins, typically operating at 70–80% of the rated values of J_c_ and B_c2_. For T_c_, the safety margin is even more critical, particularly given the anticipated broader availability of liquid hydrogen (LH_2_) in the near future (see Figure 3). Cryogenic cooling systems, whether direct (d-LH_2_) or indirect (i-LH_2_), must be engineered for thermal stability and redundancy to prevent local hot spots from escalating into system-wide failures.

While critical temperature dictates the choice of cryogen, the global pinning force versus magnetic field is arguably the most important intrinsic parameter determining the cryomagnetic applicability of a superconductor. In most high-pinning, type-II superconductors, the global pinning force, often normalized as F_p_/F_pmax_, exhibits a scaling behavior with reduced magnetic flux density b = B/Bc_2_, as shown in Figure 10 and described by Equations (1)–(3).Pinning force F_p_ = J_c_ × B(1)F_p_ = J_c_B = G(B_c2_(T))^2^ f(b)/k^m^(2)f(b) = b^p^(1 − b)^q^(3)

This scaling law is powerful: measuring F_p_ versus *b* at one temperature allows prediction of F_p_ at other temperatures using a scaling factor of the form [Bc_2_(T)]^n^. The equation includes several parameters: G is a geometrical factor associated with the microstructure (typically interpreted as the surface area of inclusions per unit volume of matrix); n and m are fitting parameters; k is a smoothing factor that defines the peak pinning value; and p and q are pinning-related and material-specific parameters. The position of the maximum pinning force along the reduced flux density axis b is given by the ratio p/(p + q).

**Figure 10 materials-18-03665-f010:**
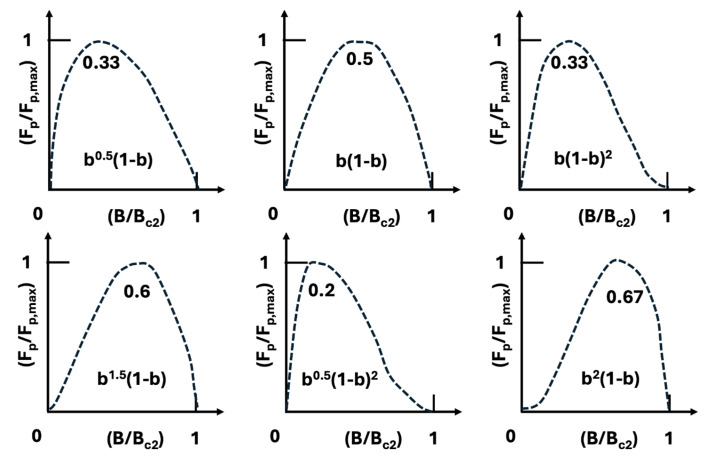
Normalized scaling law of pinning force dependence on magnetic field and temperature: The pinning force F_p_ of all superconducting materials can be expressed as a normalized function (F_p_/F_p,max_) vs. (B/B_c2_) where F_p,max_ is the maximum pinning force and B_c2_ is the upper critical magnetic field. This scaling behavior follows the formulation by Coote R.I. Reproduced with permission from R.I. Coote, Ph.D. Thesis, Department of Materials Science and Metallurgy, published by Cambridge University, Cambridge, UK, 1970 [19,20].

For many advanced superconductors considered for practical applications—such as Powder-in-Tube (PIT) Nb_3_Sn, MgB_2_, and Bi-2212—the pinning force scaling follows the form: F_p_ ~ b^0^·^5^(1 − b)^2^ where p = 0.5, q = 2, and the peak pinning occurs at b = 0.2 of B_c2_. This presents a limitation for high-field applications, especially in A15 and MgB_2_ conductors. To improve performance under higher magnetic fields at elevated temperatures (e.g., near 14 K with LH_2_ cooling), the superconductor’s pinning should instead follow the scaling form: F_p_ ~ b^2^(1 − b) where p = 2 and q = 1. This shift moves the pinning force maximum toward higher reduced fields, making the conductor more suitable for high-field operation with LH_2_-based cryogenic systems, as illustrated in Figure 11.

As shown in Figure 12, many A15 conductors demonstrate significant potential for high-field pinning performance, characterized by the scaling law F_p_ = J_c_ × B ~ b^2^(1 − b) (with p = 2, q = 1), which makes them well-suited for operation at elevated cryogenic temperatures, such as those achievable with liquid hydrogen (LH_2_). This high-field scaling behavior enables A15 materials to take full advantage of hydrogen cryomagnetic cooling regimes in the 15–30 K range.

**Figure 11 materials-18-03665-f011:**
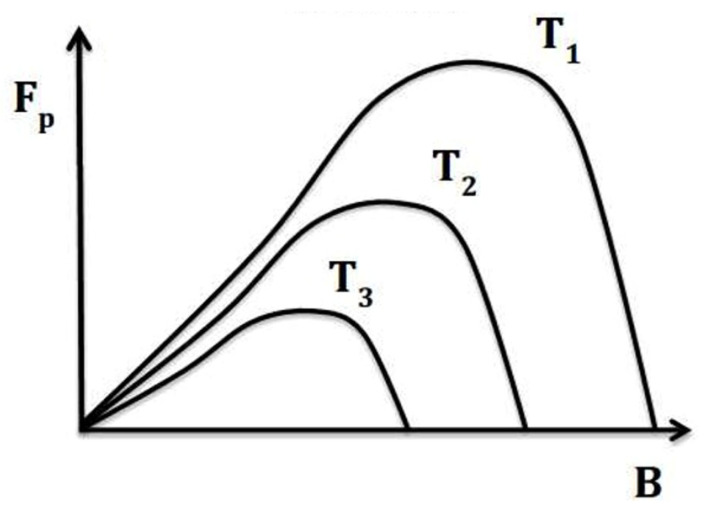
Volume pinning force F_p_ versus temperature for an exemplary A15 superconductor. The graph illustrates that the maximum pinning force occurs at high normalized magnetic flux density, which is advantageous for maintaining superior superconducting performance at elevated temperatures. The pinning force is given by F_p_(T) = J_c_ × B(T), where T_3_ > T_2_ > T_1_ [21]. Reproduced with permission from B.A. Glowacki, Acta Physica Polonica A, published by IP PAS; 2016.

For example, Nb_3_Sn conductors doped with tantalum, manufactured via a mixed-phase liquid-solid diffusion process [(Nb,Ta)_6_Sn_5_ + Nb–4at%Ta], exhibit enhanced high-field critical current characteristics. These make such conductors excellent candidates for elevated-temperature superconducting applications, as illustrated in Figure 11. Even more promising are the high-field properties achieved in single-filament Nb_3_Ga conductors, which have undergone quenching followed by additional annealing. With a critical temperature T_c_ ≈ 20 K and superior J_c_ – B characteristics at 4.2 K, these conductors show strong potential for direct LH_2_ cooling (d-LH_2_) applications, also depicted in Figure 11.

**Figure 12 materials-18-03665-f012:**
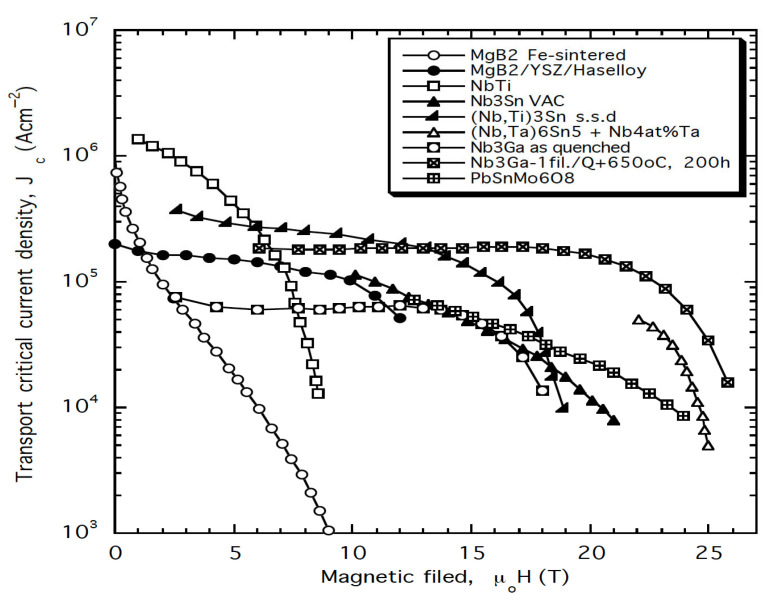
Comparison of critical current density versus external magnetic field for the range of superconducting conductors manufactured by different processes as specified in the legend, including NbTi, Nb_3_Sn, Nb_3_Ga, PbSnMo6O_8_, and MgB_2_ [22]. Comparison of critical current density J_c_ versus external magnetic flux density B for various superconducting conductors produced by different manufacturing processes [10]. Measurements conducted at 4.2 K. Reproduced with permission from B.A. Glowacki, Kirk-Othmer Encyclopedia of Chemical Technology; published by JOHN WILEY and SONS, 2005.

Furthermore, Nb_3_Ge, with a T_c_ reaching 23.2 K (see the next paragraph, Table 1, and Figure 13), offers even better performance in the LH_2_ temperature range. Its properties make it a prime candidate for high-field applications cooled by liquid hydrogen.

Given this context, the next chapter will focus on the potential of Nb_3_(Al,Ge) conductors for direct LH_2_ cooling and their role in advancing hydrogen-based cryomagnetic technologies.

### 4.1. Nb_3_(Al, Ge)

An important subgroup within the metallic A15 superconducting family includes Nb_3_Al, Nb_3_(Al,Ge), and Nb_3_Ge, which exhibit critical temperatures (T_c_) in the range of 20–23 K, and exceptionally high values of B_c2_ (Table 3). These materials enable the construction of 20 T-class magnets operating at ~14 K, as shown in Figure 13, exceeding the performance of both Nb_3_Sn and Nb_3_Ga superconductors [23,24].

**Figure 13 materials-18-03665-f013:**
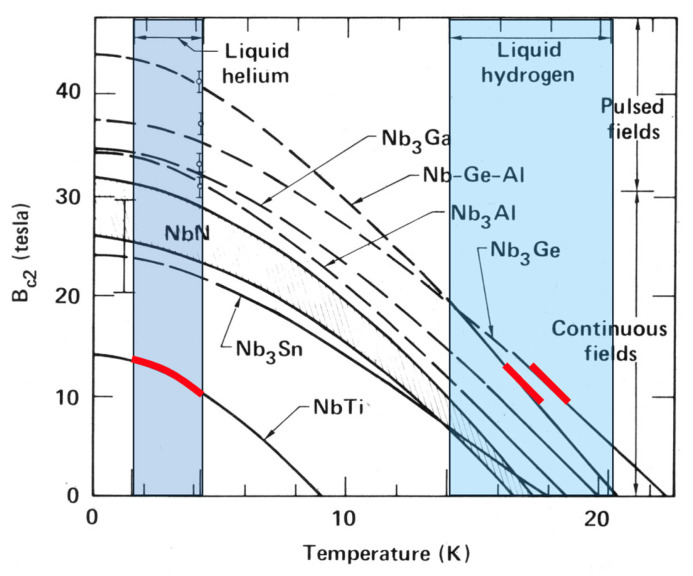
Upper critical magnetic flux density B_c2_ versus temperature for Nb-based superconducting conductors [25]. The melting temperature of hydrogen (~14 K), relevant to “slash hydrogen” used in rocket propulsion, marks a practical cooling benchmark. Nb_3_(Ge,Al) conductors can operate effectively in the 15–20 T range near this temperature [12]. The red lines highlight that NbTi wires generate magnetic fields of approximately 10 T at 4.2 K (liquid helium temperature), while Nb_3_(Al,Ge) and Nb_3_Ge conductors achieve comparable flux densities at around 18 K, a temperature accessible with dense liquid hydrogen (d-LH_2_) cooling. Adapted from Ref. [25], reproduced with permission from B.A. Glowacki, Acta Physica Polonica A; published by IP PAS; 2016.

Substitution of Al with elements such as Ge, Ga, Be, B, or Cu within the A15 structure has been shown to increase the T_c_ of Nb_3_Al. Among these, germanium is particularly effective at stabilizing the A15 phase. The most widely used fabrication technique for Nb_3_(Al,Ge) involves powder metallurgy (PM), where Nb, Al, and Ge powders are packed into a Nb sheath [24]. The composite is typically heat-treated via laser or electron-beam irradiation to suppress excessive grain growth. However, the limited cooling rate of this method results in degraded J_c_ at low magnetic fields.

A more promising method was developed by Kikuchi et al. [26], using a rapid heating, in situ transformation (RIT) process. In this technique, a Nb/Al–20%Ge composite is rapidly heated and then quenched, forming Nb_3_(Al,Ge) filaments embedded in a Nb matrix, directly yielding the A15 phase. A subsequent heat treatment at 800 °C for 10 h significantly enhances the long-range order of the A15 phase, thereby increasing T_c_. The resulting J_c_ at high fields is significantly higher than that of RHQT (rapid heating, quenching, and transformation) Nb_3_Al.

To enhance the applicability of Nb_3_(Al,Ge) conductors for NMR magnets operating with LH_2_ cooling, the J_c_ peak must be shifted to higher magnetic fields, as seen in the J_c_ vs. H plot in Figure 14a. A comparison of the volume pinning force between wires produced via rapid heating/quenching and those produced by solid-state diffusion reveals a marked difference in pinning behavior (Figure 14b). For Nb_3_(Al,Ge), the maximum volume pinning force occurs at higher magnetic fields, while in solid-state-reacted Nb_3_Al wires, the peak is observed at ~5 T.

This shift is associated with differences in pinning mechanisms. For the irradiated Nb_3_(Al,Ge) samples, the volume pinning force at lower fields is suppressed, likely due to point defects or interfacial pinning between superconducting regions of differing T_c_. Magnetic pinning at the boundaries of different phases, with dissimilar magnetization curves, can also result in a minimum in J_c_ vs. B, where the magnetizations equalize and both J_c_ and F_p_ approach zero. Between this minimum and H_c2_, a J_c_ peak appears, commonly referred to as the “valley effect”, first noted by Evetts [27]. In addition to magnetic pinning, Pippard [28] proposed an alternative mechanism related to flux lattice defects and weak, line-like pinning forces. According to his theory, the J_c_ peak tends to occur near H_c2_, as the rigidity of the flux-line lattice drops more rapidly than the pinning force exerted by inhomogeneities. This phenomenon further explains the enhanced high-field performance of Nb_3_(Al,Ge) under LH_2_ cryogenic conditions.

**Figure 14 materials-18-03665-f014:**
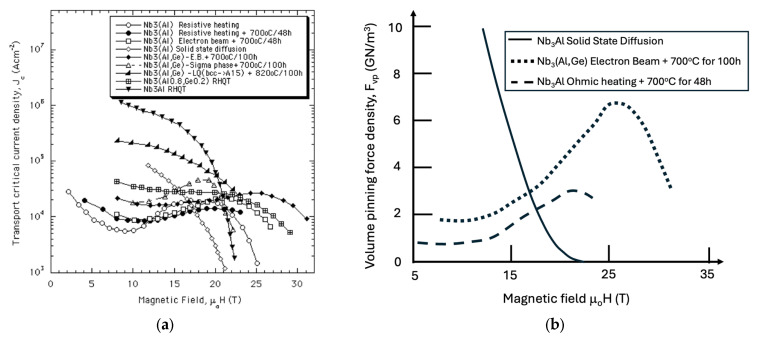
(**a**) Critical current density (J_c_) versus external magnetic field for Nb_3_Al and Nb_3_(Al,Ge) conductors produced by various manufacturing techniques [23,24,26,29,30,31,32,33,34,35,36]. (**b**) Corresponding global pinning force density (F_p_ = J_c_ × B) of round Nb_3_Al and Nb_3_(Al,Ge) conductors fabricated by rapid ohmic heating and electron beam irradiation. The F_p_ values are derived from the J_c_ vs. B data shown in panel (**a**) (measurements conducted at 4.2 K). Reproduced with permission from B.A. Glowacki, Intermetallics; published by Elsevier, 1999.

This loss of rigidity in the flux-line lattice (FLL) strengthens flux pinning by allowing fluxoids to more easily conform to a minimum energy configuration under the influence of the applied current [28]. As a result, the J_c_ peak is observed over a broader range of conditions than would be expected in a simple J_c_ vs. H relationship. This peak originates from a narrow maximum in the pinning force (F_p_) at high reduced fields. Microstructures that exhibit either a high density of weak pinning centers or a moderate density of stronger pins can generate the desired narrow F_p_ peak at high fields. A high density of defects in the FLL allows fluxoids to adapt more effectively to pinning centers, thereby enhancing J_c_ [37]. In Nb_3_Al and similar superconductors, this results in an increasing J_c_ with increasing magnetic field, up to the peak. The explanation lies in the deformation of the flux lattice, which enables better accommodation of fluxoids into the pinning sites. However, beyond the peak, depinning occurs, causing a sharp drop in J_c_ [24]. Optimizing the J_c_ peak and its position in Nb_3_(Al,Ge) remains a complex challenge due to the sensitivity of the intermetallic phase formation to thermal and mechanical processing conditions. Currently, the rapid-heating/quenching (RHQ) process is the preferred method for producing Nb_3_(Al,Ge) wires. The alternative transformation-heat-based up-quenching (TRUQ) process has not yet reached a mature stage of development [23].

Recent findings show that reducing the Al–Ge alloy core size in the precursor wire leads to improved high-field J_c_. For instance, Nb_3_(Al,Ge) wires with a 0.3 µm core achieve J_c_ > 150 A/mm^2^ at 4.2 K and 25 T, nearly twice the value of wires with 1.5 µm cores. This improvement is attributed to a higher diffusion-pair density in the composite, which increases the volume fraction of the A15-Nb_3_(Al,Ge) phase [26].

The Nb_3_(Al,Ge) system thus presents a promising alternative in high-performance superconductors, particularly for operation at 15 K, a temperature that is readily achievable with liquid hydrogen (LH_2_). This makes Nb_3_Ge-based conductors particularly suitable for hydrogen cryomagnetic applications. A schematic performance-to-energy cost evaluation, defined as J_c_ per kW of cryogenic cooling, is shown in Figure 15, expanding on the principle introduced in Figure 9a. The analysis indicates that the optimal performance-to-cost ratio occurs at 14–15 K using either direct (d-LH_2_) or indirect (i-LH_2_) cooling, thereby affirming the practical suitability of these materials for next-generation hydrogen-cooled superconducting systems.

**Figure 15 materials-18-03665-f015:**
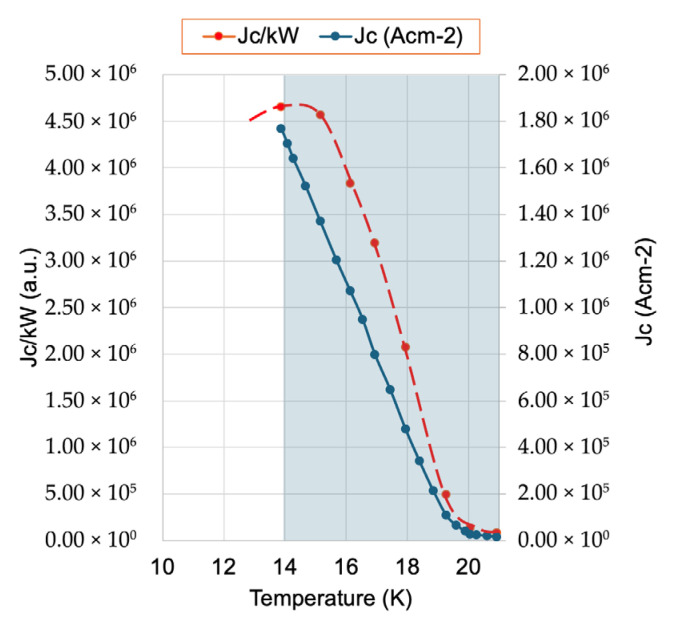
Dependence of the critical current density normalized by cooling power (J_c_/kW) as a function of temperature in the range 13 K to 21 K, alongside the corresponding critical current density (J_c_) versus temperature in the range 14 K to 21 K, based on data from [38]. The figure follows the formatting conventions of Figure 9a and specifically illustrates performance characteristics for Nb_3_Ge. The analysis reveals an optimal operational region around 15 K, where superconducting performance (J_c_) remains high while the cooling power requirement remains economically favourable. The shaded blue area indicates the temperature range where hydrogen exists in its liquid phase under atmospheric pressure, between its melting and boiling points, emphasizing the practical relevance of liquid hydrogen cooling for intermediate-temperature superconductors such as Nb_3_Ge.

While Nb_3_(Al,Ge) A15 conductors do not achieve the high critical temperatures of REBCO or cuprate superconductors such as Hg-1223 and Tl-1223, they offer significant advantages in terms of material safety, sustainability, and scalability. Nevertheless, as with all advanced materials, issues related to resource availability and geopolitical concentration must be carefully considered.

A key benefit of Nb_3_Ge lies in its composition: it is based on non-toxic, relatively abundant elements. Unlike mercury- or thallium-based high-temperature superconductors, which pose serious environmental and health hazards, Nb_3_Ge is composed of niobium and germanium, both widely used in high-tech industries and much safer to process and handle. This makes Nb_3_Ge particularly attractive for large-scale or long-term applications, where material toxicity and environmental impact during manufacturing and disposal are of concern. Although Nb_3_Ge requires cooling to lower temperatures, the use of liquid hydrogen (LH_2_) presents fewer environmental risks compared to the toxic components in some high-temperature superconductors.

In terms of reliability and long-term sustainability, Nb_3_Ge also compares favorably with rare-earth-based superconductors such as YBCO, which rely on elements with fragile and geopolitically constrained supply chains. The global supply of rare-earth elements is often dominated by a few countries, subject to price volatility and export restrictions. In contrast, niobium and germanium are established industrial materials with broader, though still concentrated, global usage. Their extraction processes do not depend on environmentally intensive RE mining, making Nb_3_Ge less exposed to the geopolitical risks that can affect REBCO scaling.

However, the concentration of production for both niobium and germanium cannot be ignored. In 2023, global germanium production was approximately 118 metric tons, with China accounting for over 93.5% of that total. Meanwhile, niobium production stood at around 83,000 metric tons, with Brazil responsible for approximately 90% (75,000 tons). These statistics underscore China’s dominance in germanium and Brazil’s leadership in niobium, creating potential vulnerabilities. Although neither element is currently classified as critically scarce, disruptions in trade policies, export controls, or political instability in these regions could adversely affect the long-term scalability of Nb_3_Ge-based superconducting systems.

From a performance perspective, both Nb_3_Ge and Nb_3_(Al,Ge) conductors exhibit strong critical current densities (J_c_) and robust magnetic field tolerance at 15 K, positioning them as excellent candidates for high-field applications such as fusion energy devices, advanced power systems, and next-generation MRI or NMR magnets. While liquid hydrogen cooling is more costly than liquid nitrogen, Nb_3_Ge’s stability and performance retention at i-LH_2_ temperatures ensure a reliable superconducting state with minimal degradation over time. This makes Nb_3_Ge particularly compelling in systems where high magnetic field performance is more important than maximizing the operating temperature.

A typical MRI system uses a GM or pulse-tube cryocooler to operate in “zero boil-off” mode or via direct conduction, particularly in mobile or stationary installations. During scanning, the cryocooler is often turned off to minimize acoustic and electromagnetic noise. In a hydrogen-based cryomagnetic infrastructure, liquid hydrogen (LH_2_) can serve simultaneously as a coolant, energy store, and energy source, offering additional hospital-wide benefits through cryogenic integration. Existing MRI facilities already include helium exhaust ducts for quench events; these can be repurposed to accommodate compressed helium close-circuit lines, with precooling via LH_2_ heat exchange, and optionally, final cooling using a Joule–Thomson (JT) stage. Such configurations enable efficient operation in the 4–14 K range for NbTi-based MRI systems, enhancing the sustainability and resilience of clinical cryogenics.

### 4.2. MgB_2_ Conductors

A substantial research effort has been devoted to the doping and development of binary MgB_2_ conductors, with the objective of enhancing their critical parameters to meet the requirements of applications at hydrogen cryogenic temperatures. Current research in MgB_2_ focuses on improving three fundamental superconducting parameters: the critical temperature (T_c_), the upper critical flux density (B_c2_), and consequently, the critical current density (J_c_) as a function of magnetic flux density (B), as illustrated in Figure 8. Among various conductor architectures, powder-in-tube (PIT) round conductors have emerged as the preferred choice due to their capability for low AC losses and their potential to generate high magnetic flux densities under hydrogen cryomagnetic conditions [39]. To fully realise MgB_2_’s potential as a competitive superconducting material, two parallel development paths are required: thin-film fabrication [40] and PIT processing [41,42].

Initial comparisons between PIT and thin-film MgB_2_ conductors have shown the superior pinning performance of thin films, particularly in terms of scaling law behavior and absolute pinning force [40], as shown in Figure 16. 

Future advancements in PIT conductor design are expected to follow a similar processing procedure to that of the A15 Nb_3_Sn multiphasic composite (e.g., Nb–Cu–NbSn_2_), where multistage liquid-phase diffusion processing led to substantial improvements in wire performance [46]. These A15 conductors are noted for their fine-grained microstructure with a high grain boundary density, resulting in exceptionally high J_c_ values and enhanced pinning strength. Analogously, the manufacture of PIT MgB_2_ conductors should evolve toward a continuous production process [47], integrating rapid heating techniques and offering flexibility for either on-line or batch-mode thermal treatments [34].

An extensive development in PIT technology has led to notable improvements in J_c_ vs. B performance for round wires, depicted in Figure 17. 

For instance, SiC-doped Cu-sheathed PIT conductors, fabricated via hot isostatic pressing (HIP), have demonstrated high J_c_ values in the 7–14 T range [50]. While HIP introduces limitations in scalability due to its batch-processing nature, it nonetheless represents a critical advancement, offering a potential pathway for PIT conductors to eventually outperform thin films in applications related to hydrogen cryomagnetic, including DC, AC, and transient current (charging/discharging) systems.

The enhanced high-field pinning behavior observed in these advanced MgB_2_ conductors appears to result from a complex interplay of powder densification, liquid–solid interfacial reactions, chemical substitution, and precipitation dynamics, often under conditions of non-uniform magnesium excess or deficiency. Ongoing research aims to characterize this behavior through model conductors prepared under varied compressive processing techniques, seeking to clarify the mechanisms responsible for forming dense, Cu-sheathed, ex situ/in situ hybrid single-core wires. Given the competing demands of densification and grain size control, non-linear constrained optimization methods are necessary to develop optimum sintering and reaction profiles, balancing MgB_2_ formation with minimal grain coarsening. The volume pinning forces achieved in high-performance MgB_2_ wires conform to a scaling law of the form ~ b^2^(1 − b), which is characteristic of robust high-field superconductors. This behavior ensures that such conductors will perform efficiently at elevated cryogenic temperatures, particularly within the 14–20 K range typical of liquid hydrogen cooling. MgB_2_ PIT conductors, therefore, represent a promising candidate for hydrogen-cooled superconducting magnets, capable of delivering high performance for a wide range of electromagnetic applications operating around 15 K, where both material efficiency and cooling economy are optimized.

## 5. High Temperature Superconductors

Over the past 38 years, a substantial number of cuprate high-temperature superconductors (HTSs) have been discovered, as illustrated in Figure 6, with critical temperatures reaching up to 135 K at ambient pressure [51]. Among them, HgBa_2_Ca_2_Cu_3_O_8_ + δ (Hg-1223) and TlBa_2_Ca_2_Cu_3_O_8_ + δ (Tl-1223) stand out for their exceptional superconducting properties, particularly at elevated temperatures. However, despite their impressive performance, including operation at liquefied natural gas (LNG) temperatures around 112 K, their widespread application is severely limited by the toxicity and environmental hazards associated with mercury (Hg) and thallium (Tl); see Table 1.

The synthesis of Hg-1223 and Tl-1223 involves high-temperature reactions requiring toxic precursors and specialized containment systems. Mercury vapor and thallium fumes pose serious safety and environmental risks due to their volatility and difficulty in containment. Without rigorous safeguards, these toxic species can contaminate air and water supplies. Consequently, regulatory authorities, particularly in the EU, have imposed strict limitations or outright bans on the production and commercial use of such materials. These safety and environmental considerations, outlined in Table 1, make Hg-1223 and Tl-1223 impractical for real-world superconducting technologies, despite their superior intrinsic properties.

Similar concerns extend to arsenic-based iron pnictide superconductors and lead-based cuprates (e.g., Pb-2223). While some of these materials exhibit promising superconducting characteristics, health and environmental risks, combined with complex processing requirements, will likely prevent them from achieving commercial viability soon.

In this chapter, we do not aim to glorify widely recognized HTS materials such as Bi_2_Sr_2_CaCu_2_O_8_ (Bi-2212), Bi_2_Sr_2_Ca_2_Cu_3_O_10_ (Bi-2223), and REBCO (rare-earth barium copper oxides), despite their notable performance at cryogenic temperatures between 64 K and 77 K, as seen in Figure 6. The reality is that these materials face numerous technical and economic challenges. These include the following:Dependence on rare-earth elements with concentrated and geopolitically sensitive supply chains.Relatively low irreversibility fields, which limit performance under high magnetic fields.Extremely complex and costly manufacturing processes, especially for REBCO tapes;Lack of scalable, wide-tape production infrastructure.Absence of multifilamentary or transposed technologically scalable configurations, which are essential for AC and dynamic current applications.And above all, their prohibitively high cost, which restricts their accessibility and scalability.

Given these limitations, the practical deployment of HTS materials remains constrained, and in many cases, low-temperature superconductors (LTSs) such as Nb_3_Sn, Nb_3_(Al,Ge), and MgB_2_, particularly those operable with liquid hydrogen at 15–20 K, may offer more viable pathways for sustainable, large-scale applications in energy, transport, and magnet technologies.

Among the various high-temperature superconductors (HTSs) developed over the past four decades, only the bismuth-based cuprates (Bi-2223 and Bi-2212) and the rare-earth barium copper oxides (REBCO or RE-123) have retained technological relevance [52]. However, each comes with significant limitations that hinder their broad commercial deployment. For Bi-based cuprates, although they exhibit sufficient performance under magnetic fields at liquid hydrogen temperatures (see Figure 18), they suffer from a low irreversibility field at higher temperatures, making them unsuitable for many practical applications. Despite the mature manufacturing processes for Bi-2223 and steady improvements in its performance, its technological reproducibility for AC applications remains poor. This is primarily due to retrograde densification, which induces intrafilamentary bridging and coupling losses, rendering AC applications inefficient and unreliable. These issues have led leading manufacturers, such as Sumitomo Electric Industries, to cease production, deeming it economically unsustainable. Nevertheless, Bi-2212 and Bi-2223 remain useful in niche DC high-field magnet applications (Figure 19a) [53]. However, their limited market applicability does not justify expectations for economic scalability or widespread adoption.

Several technological challenges persist in HTS coil design, particularly for Bi-based and REBCO systems: Screening-current-induced magnetic fields, which distort the field profile; Mechanical degradation, due to high stresses in both longitudinal and transverse directions and Insufficient quench protection, as abrupt thermal runaway events are difficult to detect and manage [54]. While DC applications can tolerate flux creep to a certain degree, AC applications, such as power transmission cables, rotating machinery (motors/generators), fusion magnet systems, and superconducting magnetic energy storage (SMES), are fundamentally limited by the high anisotropy and 2D architecture of thin-film REBCO conductors. Multifilamentary structuring and transposition, which are essential for minimizing AC losses, are not currently feasible at scale with REBCO tapes.

Despite over 38 years of intensive global R&D since the 1987 discovery of REBCO, key challenges remain unresolved. Notably, the development of wide, multifilamentary REBCO conductors continues to lag. A comprehensive review [55] highlights the urgent need for the following:Optimizing pinning defect structures at 20–40 K, especially under variable flux density conditions (Figure 19a).Tailored pinning architectures, which are not yet achievable with current fabrication technologies.And the development of low-cost, scalable, and highly reliable manufacturing techniques to replace today’s expensive and limited 2D coated conductor architecture.

In contrast, metallic superconductors, such as advanced A-15 (e.g., Nb_3_Al, Nb_3_Ge) and MgB_2_, are already operable at liquid hydrogen temperatures (14–20 K), offering higher performance at lower cost, and far greater feasibility for AC and dynamic current applications. While REBCO conductors can be operated at higher temperatures (64–77 K, using liquid nitrogen), this offers diminishing value in comparison to the superior performance achievable at 20 K, see Figure 20b, with hydrogen-based cryogenics, which are increasingly accessible in the hydrogen economy.

Commercially available REBCO tapes, typically 6 mm wide, can sustain high persistent currents within their micron-thick superconducting layers. However, these narrow tapes are not ideal for persistent magnet applications, and their production is costly and time consuming. An innovative alternative involves cutting wider (e.g., 40 mm) REBCO tapes into smaller segments, stacking them to form composite bulk magnets, Figure 20a. These stacked tapes can trap exceptionally high magnetic fields at cryogenic temperatures, despite containing only a small volume fraction of REBCO material.

A remarkable demonstration of this approach is the trapping of a 17.7 T magnetic field at 8 K between two REBCO tape stacks, with no external mechanical reinforcement. Even at 14 K, a field of 17.6 T could be maintained. This represents the strongest permanent magnet ever reported for superconducting tapes and showcases the potential of REBCO stacked tapes for persistent cryogenic hydrogen magnet applications (see Figure 20, Figure 21, Figure 22 and Figure 23). In these configurations, REBCO tape stacks outperform conventional (RE)BCO bulks in both mechanical robustness and magnetic field strength.

**Figure 20 materials-18-03665-f020:**
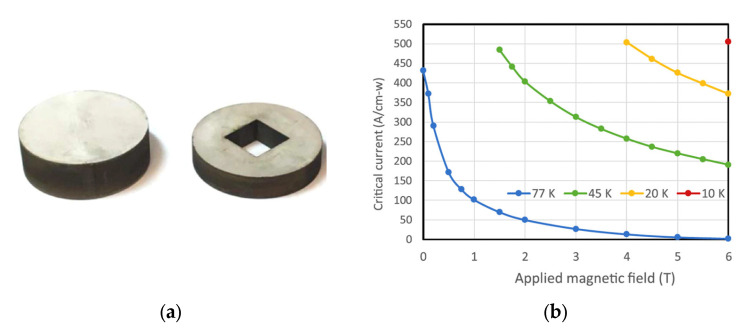
(**a**) Components of one half of the American Superconductor (AMSC) stack after spark erosion machining, showing 129 layers on one side and 79 layers on the other. The stack has an external diameter of 34.4 mm [43]. (**b**) Critical current data (A/cm-width) for the AMSC (RE)BCO tape used in the hybrid stack of HTS tapes. The performance at 20 K, enabled by liquid hydrogen (LH_2_) as an energy carrier and cryogen, is clearly superior to that at 77 K with liquid nitrogen (LN_2_) [56]. Reprinted from Ref. [56].

Historically, most trapped field magnets have employed bulk (RE)BCO superconductors grown via top-seeded melt growth methods [57,58], achieving high HTS volume fractions (>90%). This high superconducting content enables exceptionally high engineering current densities. However, the intrinsic brittleness of the ceramic material limits its mechanical performance. Cracking due to tensile stresses, induced by Lorentz forces acting between circulating supercurrents and the pinned magnetic flux vortices, presents a major challenge. Furthermore, the ability of the critical current to flow along the c-axis in bulk samples complicates flux pinning mechanisms and increases the risk of magnetic flux instabilities, particularly at low temperatures.

Recent comparative studies have shown that REBCO tape stacks, especially those made from 12 mm wide tapes, offer significant advantages over bulk HTS, as presented in Figure 21. As the operating temperature decreases from 50 K to 10 K, the trapped field in stacks improves by nearly 40%, while bulk samples plateau or degrade due to their mechanical fragility. Crucially, liquid hydrogen (LH_2_) operation offers a thermal environment where stacked REBCO conductors outperform in both mechanical and magnetic field performance. This validates LH_2_ as the cryogen of choice for next-generation cryomagnetic systems.

**Figure 21 materials-18-03665-f021:**
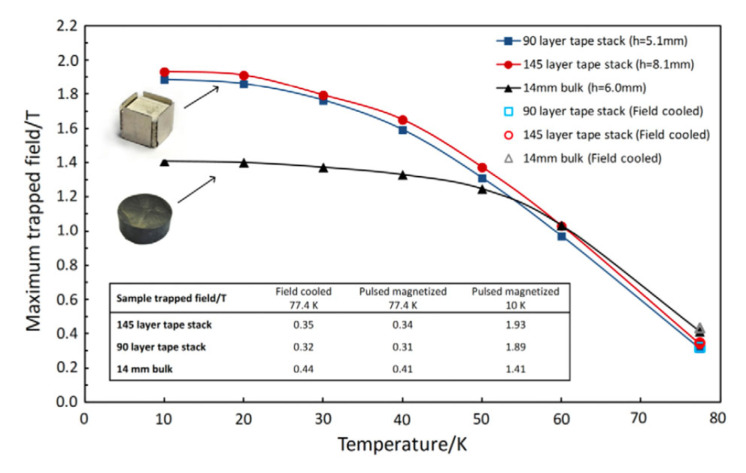
Peak trapped magnetic field measured 0.8 mm above tape and bulk samples following pulse field magnetization as temperature decreases. Magnetization was performed using multiple pulses at each temperature, starting from 77.4 K and progressing downward. Results for field cooling in an electromagnet at 77.4 K are also shown (hollow marker) [59]. Reproduced with permission from B.A. Glowacki, sUPER. sCI. and Technol.; published by IOP, 2013.

Stacked REBCO tapes are now under serious consideration as rotor poles in superconducting electric motors, where the cryogenic energy cost is offset by the gains in power-to-weight ratio (see Figure 22 and Figure 23). The advancement of wide tape production (40–100 mm) is critical not only for reducing the cost of standard 4–6 mm REBCO tapes by slitting wide tapes, but also for enabling scalable applications in AC motors, HVDC/AC transmission cables, fusion reactor magnets, and SMES systems.

**Figure 22 materials-18-03665-f022:**
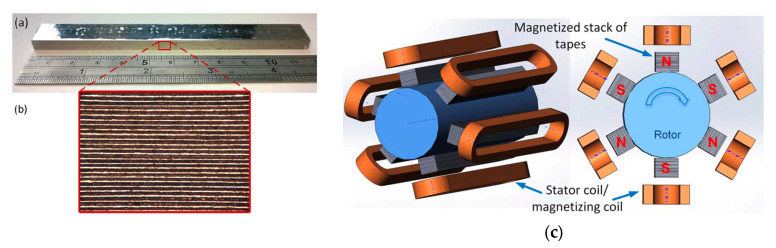
(**a**) Stack measuring 12 mm × 120 mm × 5.2 mm composed of 60 layers of superconducting tape, bonded by heating and compressing layers precoated with solder; (**b**) zoomed side-view image showing the uniformity of the layered structure within the stack; (**c**) conceptual illustration of a motor design utilizing rectangular stacks of tapes as rotor field poles, magnetized by racetrack copper coils on the stator, which also serve as the motor’s operational stator coils [60]. Reproduced with permission from B.A. Glowacki, IEEE Trans. on Appl. Supercon.; published by IEEE, 2015.

The rectangular geometry of stacked REBCO magnets aligns well with the design of radial-gap synchronous motors, inspiring new HTS motor concepts. For example, a trapped field of 0.72 T has been demonstrated in a compact 24 mm × 12 mm stack of only 15 layers, operating at 15 K (LH_2_).

**Figure 23 materials-18-03665-f023:**
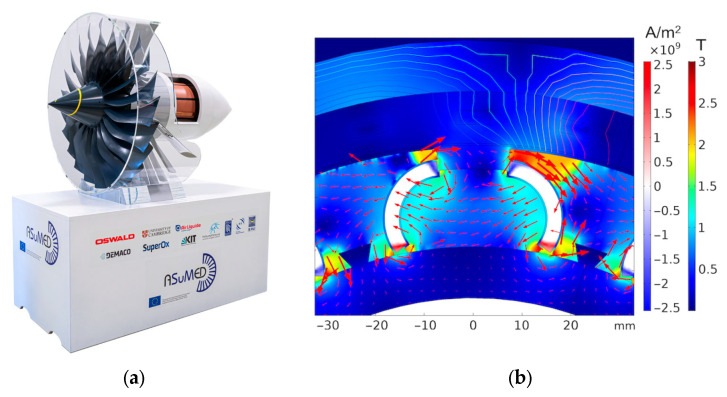
(**a**) Mockup of the 1 MW fully superconducting ASuMED axial flux motor showcased at the 2019 Hannover Fair [61]; (**b**) Current distribution in the patented-shaped REBCO stacks and corresponding magnetic flux density in the motor cross-section during operation. Nominal currents flow in the stator windings, the rotor rotates, and torque is produced [61]. Reprinted from Ref. [61].

One of the most ambitious applications is the European Union-funded ASuMED project (Advanced Superconducting Motor Experimental Demonstrator), which targets the Flightpath 2050 objectives for electrified aviation [61], Figure 23a. This project aims to develop a 1 MW all-superconducting motor with a power density of 20 kW kg^−1^, incorporating a 4 mm REBCO tape wound stator (SUPEROX) and rotor stacks of 40 mm × 60 mm REBCO tape, fabricated in collaboration with Deutsche Nano BSF (Figure 23b).

A high-efficiency cryostat system has been designed for the motor, featuring an integrated cryogenic cooling loop and power converter. The rotor is cooled indirectly using helium gas at 25 K and 2 bar with the HTS stacks operating at 30 K. Importantly, indirect LH_2_ cooling (i-LH_2_) serves as the ultimate thermal reservoir. Studies have also examined the feasibility of direct LH_2_ cooling, which could reduce the total mass of the cryogenic system and enhance system integration, especially when LH_2_ is also used as the energy carrier to power the superconducting propulsion motor.

## 6. Conclusions

Hydrogen cryomagnetic technology represents an innovative convergence of two advanced scientific domains: superconductivity and cryogenic hydrogen systems. It leverages both metallic and ceramic superconductors—materials that operate efficiently in liquid hydrogen environments. In this context, liquid hydrogen plays a dual role: it functions as a clean energy carrier and as a cryogenic coolant that maintains superconductors at their critical low temperatures. By exploiting hydrogen’s low boiling point (14–32 K), this approach reduces reliance on more costly and complex cooling systems such as those using liquid helium. However, during early-stage implementation, indirect cooling (i-LH_2_) using a closed helium compression loop with an LH_2_ heat exchanger may offer a more practical solution. The integration of superconductors with liquid hydrogen enables the development of ultra-efficient magnetic systems, including superconducting magnetic energy storage (SMES), magnetic levitation for transportation, high-field magnets for research and medical imaging, and magnetic confinement for fusion energy applications.

Beyond large-scale infrastructure, hydrogen cryomagnetic systems hold promise for decentralized energy solutions, supporting compact, lightweight, and high-capacity systems suitable for off-grid and remote locations. The dual role of hydrogen—as both fuel and refrigerant—simplifies system architecture and boosts energy density per unit of mass and volume. Ultimately, this emerging field seeks to combine the quantum-level performance of practical superconductors with hydrogen-based sustainability, offering a transformative pathway to address the critical challenges of clean energy storage, transmission, and propulsion across all superconducting technologies.

## Figures and Tables

**Figure 1 materials-18-03665-f001:**
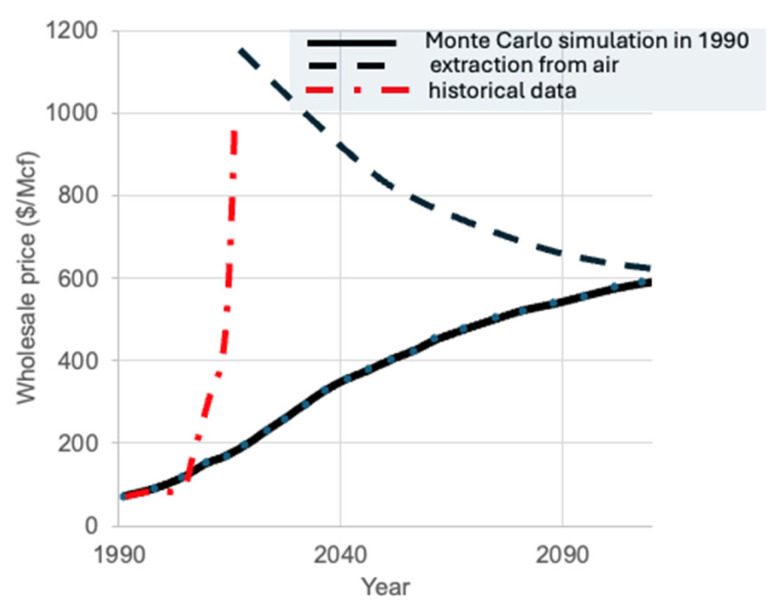
Grade A helium wholesale market price in dollars per thousand cubic feet ($/Mcf) showing scenario of 1990 Monte Carlo simulation (solid line) [11]. The red dotted-dashed line represents historical helium price data, while the broken-dashed line shows the estimated cost of helium extraction from air. Wholesale helium prices vary widely depending on buyer and helium grade, generally ranging from USD 400 to over USD 1000 per Mcf. For reference, the US Defence Logistics Agency evaluated bulk helium pricing at USD 1080/Mcf in 2023, and NASA signed a five-year supply contract at USD 918/Mcf in November 2022. Historically, liquid helium (LHe) prices at the Cavendish Laboratory were approximately GBP 2 per litre in 1990, rising sharply to around GBP 45 per litre in 2025.

**Figure 2 materials-18-03665-f002:**
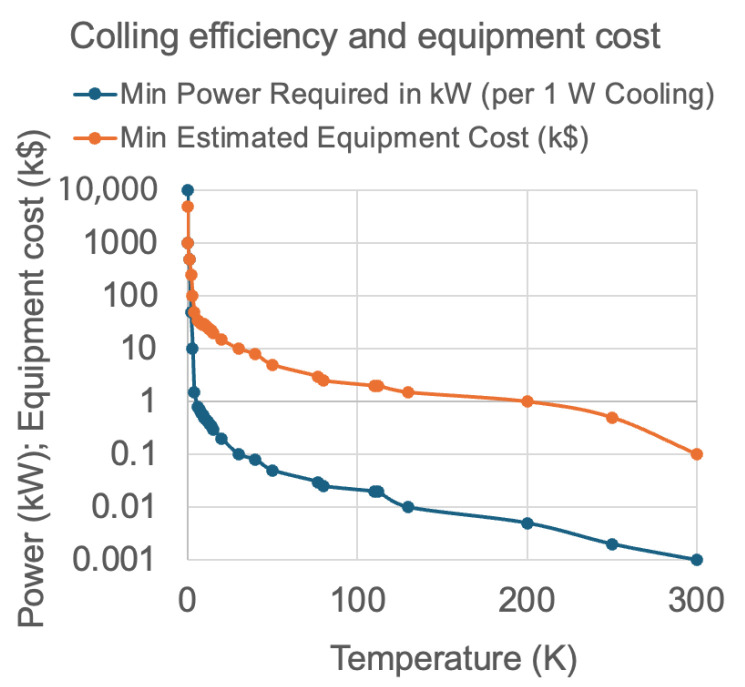
Minimum power required by cooling machinery at room temperature (in kW) to achieve 1 W of cooling power over a temperature range from 10 mK to 300 K. The graph also associates the minimum estimated equipment cost and the corresponding cooling technology system description. The cooling technologies shown include Nuclear Adiabatic Demagnetisation Refrigerator (NADR), Dilution Refrigerator (DR), Adiabatic Refrigerator (ADR), Joule-Thomson (JT) valve, Pulse Tube, Gifford-McMahon Cryocooler (GM), Stirling Cryocooler, and Thermoelectric Cooler (TEC).

**Figure 3 materials-18-03665-f003:**
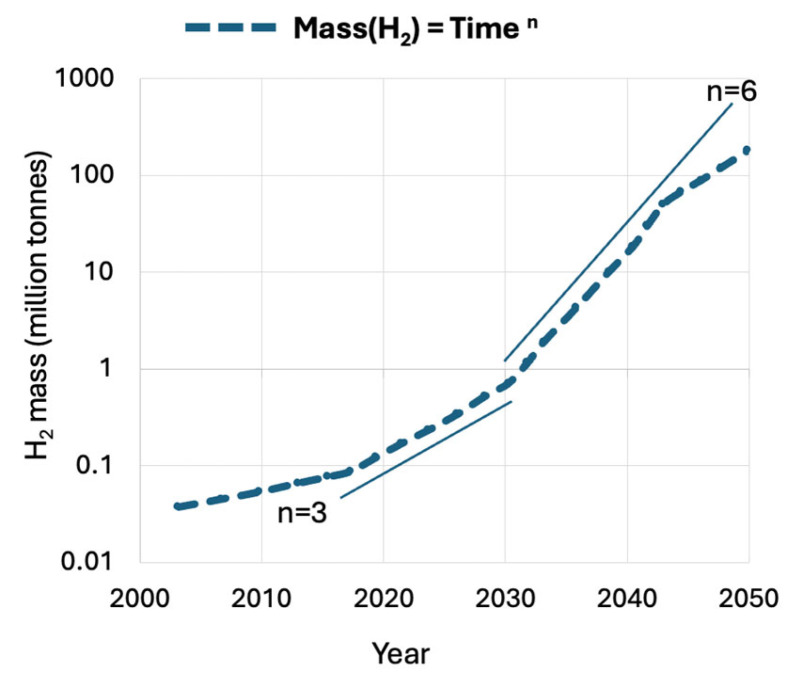
Predicted mass production of hydrogen demanded by the USA economy over time. Demand scenario is driven by greenhouse gas mitigation efforts. The hydrogen demand projections remain nearly identical up to 2050 [11]. Meanwhile, the exponent *n* in the hydrogen mass growth equation, Mass(H_2_) *=* Time ^n^, is expected to increase from n = 2 to n = 6 or higher over the next decade, indicating rapid growth in hydrogen-related demand. After 2045, it is predicted that the market will stabilize at n = 3.

**Figure 4 materials-18-03665-f004:**
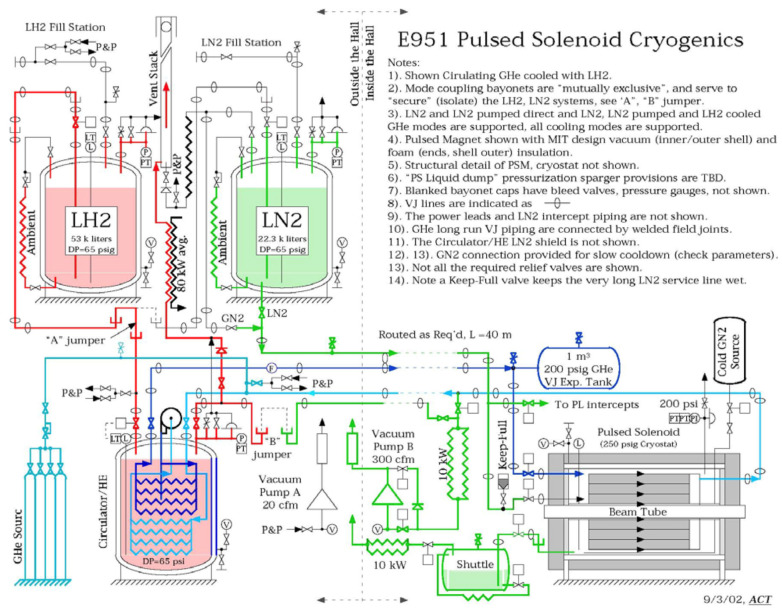
Piping and instrumentation diagram of a proposed cryogenic cooling system for a 15-T pulsed copper solenoid magnet operating at 30 K [14]. The vertical grey broken line outlines the hydrogen safety zone, marking the boundary outside the building to ensure safe handling of cryogenic hydrogen. Reproduced with permission from B.A. Glowacki, IEEE Transactions on Applied Superconductivity; published by IEEE, 2013.

**Figure 5 materials-18-03665-f005:**
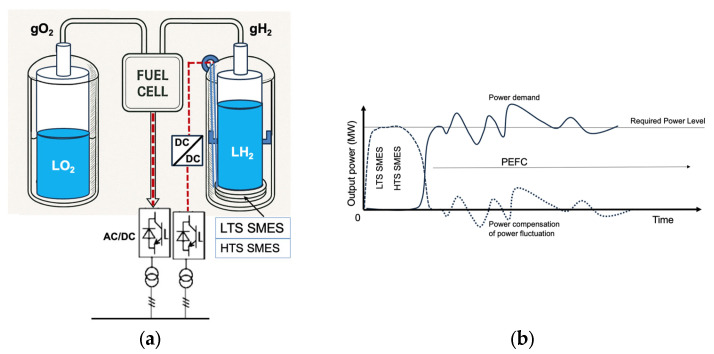
Configuration of the liquid hydrogen (LH_2_) emergency power supply system: (**a**) The installation comprises a liquid hydrogen-cooled superconducting magnetic energy storage (SMES) system with indirect LH_2_ cooling (i-LH_2_), fuel cell units, liquid hydrogen and oxygen reservoirs, and AC/DC converters [5]. Hydrogen and oxygen gases from the reservoirs feed the fuel cells. The superconducting magnet is positioned directly on the internal dewar wall, maintaining thermal contact with the LH_2_ coolant. (**b**) Schematic of emergency power sharing between SMES and the fuel cell. At blackout onset, SMES provides electric power for the first minute (broken line) while the fuel cell supplies power for the subsequent ten hours (solid line). During normal operation, SMES smooths power demand fluctuations. The superconducting material used can be MgB_2_, REBCO, Bi-2212, or Bi-2223.

**Figure 6 materials-18-03665-f006:**
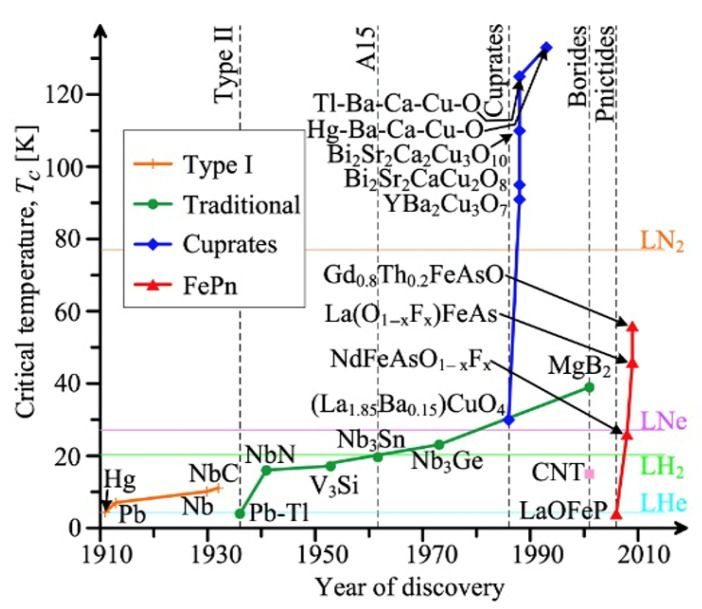
Evolution of superconducting materials over time: Critical temperature (Tc) plotted against the year of discovery. The diagram encompasses elemental (metallic) superconductors, alloy superconductors, intermetallic A-15 compounds (LTSs), intermetallic A1B2 MgB_2_, high-temperature ceramic cuprates, iron-based superconductors, and hydride-based systems under extreme pressure (HTS). Horizontal lines indicate the boiling points of key cryogenic coolants, liquid helium (4.2 K), liquid hydrogen (20.3 K), liquid neon (27.1 K), and liquid nitrogen (77 K), marking practical operating temperature thresholds.

**Figure 7 materials-18-03665-f007:**
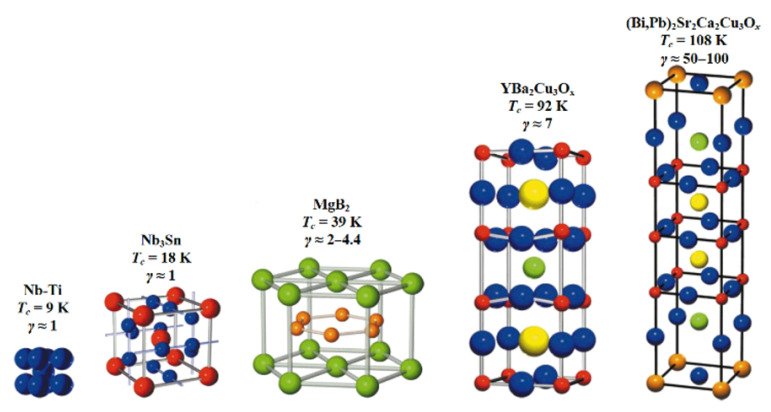
Images of crystallographic structures showcasing metallic superconductors NbTi (bcc crystal structure), Nb_3_Sn (A15 crystal structure), and MgB_2_ (hexagonal AlB2-type crystal structure) alongside oxide ceramic superconductors YBa_2_Cu_3_O_7−x_ (layered perovskite-like structure) and (Bi,Pb)_2_Sr_2_Ca_2_Cu_3_O_x_ (Bi-2223) (layered perovskite-like structure with ‘superconducting’ blocks and ‘charge reservoir’ block) [17]. Individual superconducting compound T_c_ value as well as structure-induced superconductive anisotropy, γ value are presented.

**Figure 8 materials-18-03665-f008:**
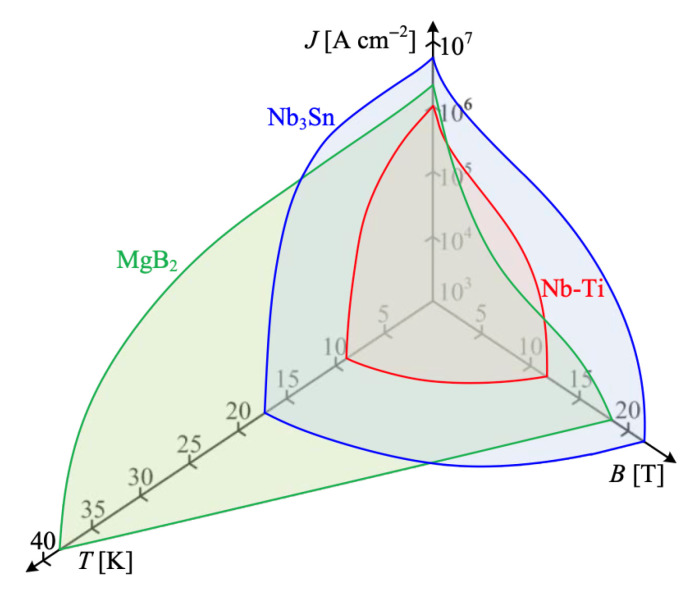
Schematic representation of the critical surface of a superconducting material characterized by three critical parameters. The current density–temperature–magnetic flux density ‘critical surface’ for commercially established practical metallic superconductors: NbTi, Nb_3_Sn, and MgB_2_ [18].

**Figure 9 materials-18-03665-f009:**
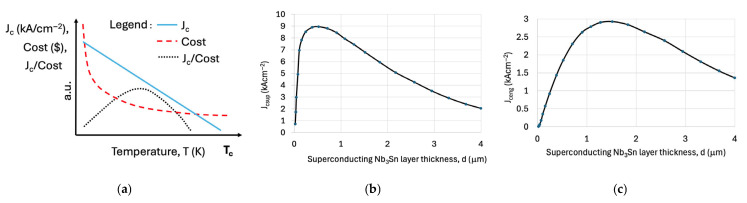
(**a**) Schematic illustrating the interrelation between superconducting critical current density, J_csup_ (T), where J_c_ can also denote engineering current density, J_ceng_, and the cooling cost of superconductors expressed in kW or dollars, as shown in Figure 2. (**b**) Example of the dependence of J_csup_ (measured at 4.2 K) on superconducting layer thickness in Nb_3_Sn tape conductor formed by liquid diffusion on NbZr 1.5% substrates. The initial zero J_csup_ corresponds to incomplete A-15 phase formation; the subsequent peak at ~0.5 μm reflects optimal grain formation, while the decrease at larger thicknesses indicates grain coarsening during extended diffusion. (**c**) Example dependence of the engineering critical current density J_ceng_ on Nb_3_Sn layer thickness in tape conductor. Variations in peak position and amplitude depend on specific material formation and deposition processes and must be considered when assessing J_ceng_ for hydrogen cryomagnetic applications.

**Figure 16 materials-18-03665-f016:**
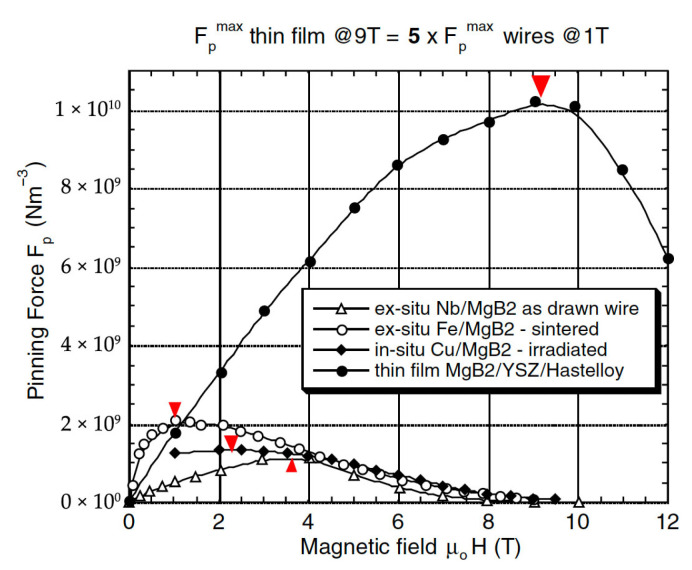
Volume pinning force curves for representative MgB_2_ conductors. The thin film MgB_2_/YSZ/Hastelloy exhibits an upper critical irreversibility field H_irr_(0) = 21 T with a maximum normalized pinning force b_max_ ~ 0.428 [40]. The PIT in situ Cu/MgB_2_ conductor irradiated by heavy ions shows H_irr_(0) = 15 T and b_max_ ~ 0.14. The PIT ex situ Fe/MgB_2_ monocore conductor has H_irr_(0) = 15 T and a lower b_max_ ~ 0.06 [43,44,45]. Reproduced with permission from B.A. Glowacki, Super. Sci. and Technol.; published by Institute of Physics, 2003.

**Figure 17 materials-18-03665-f017:**
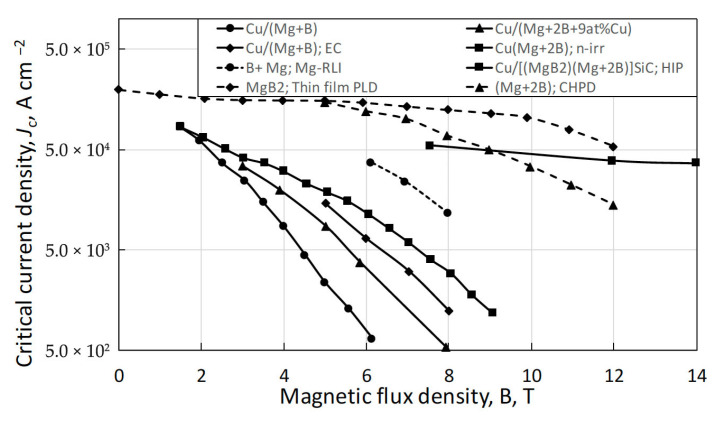
Comparison of transport critical current density J_c_ versus magnetic flux density at 4.2 K for Cu-based MgB_2_ wire technologies (solid lines) against other PIT and thin film MgB_2_ conductors (broken line solid diamonds). The Cu/[MgB_2_/Mg + 2B] wire produced by hot isostatic pressing (HIP) (solid line solid squares) achieves J_c_ values comparable to high-performance MgB_2_ thin films [40] and PIT-CHPD wires [48] at higher magnetic fields, outperforming liquid Mg reactive infiltration (Mg-RLI) wires [49]. Reproduced with permission from B.A. Glowacki, IEEE Proceedings; published by IEEE, 2017.

**Figure 18 materials-18-03665-f018:**
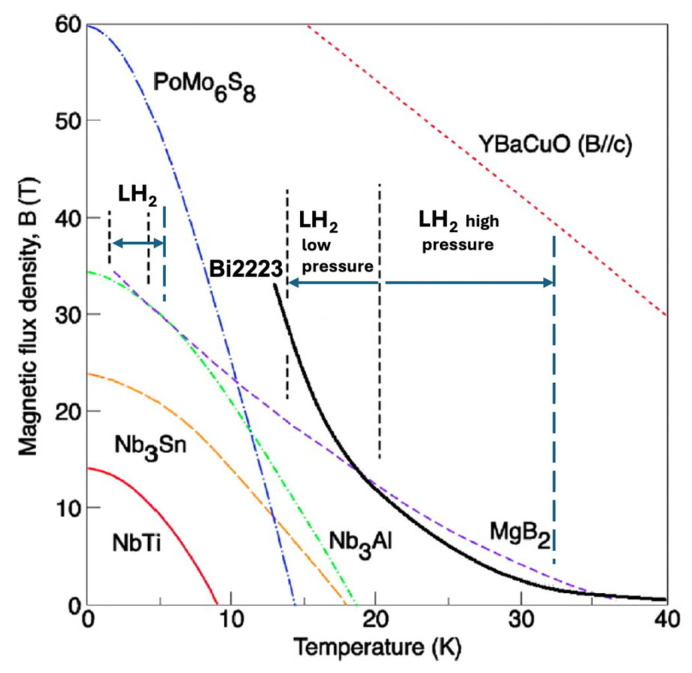
Magnetic flux density versus cryogenic temperature for selected low-temperature and high-temperature superconductors. REBCO conductors exhibit excellent performance at around 20 K, making them suitable for liquid hydrogen (LH_2_) cryomagnetic applications at this temperature. Bi-2223 and Bi-2212 conductors are viable LH_2_ cryomagnetic materials only below 20 K. MgB_2_ shows comparable or superior advantages over Bi-based compounds at these temperatures, indicating no clear benefit in using Bi-compounds instead of MgB_2_ for LH_2_ applications.

**Figure 19 materials-18-03665-f019:**
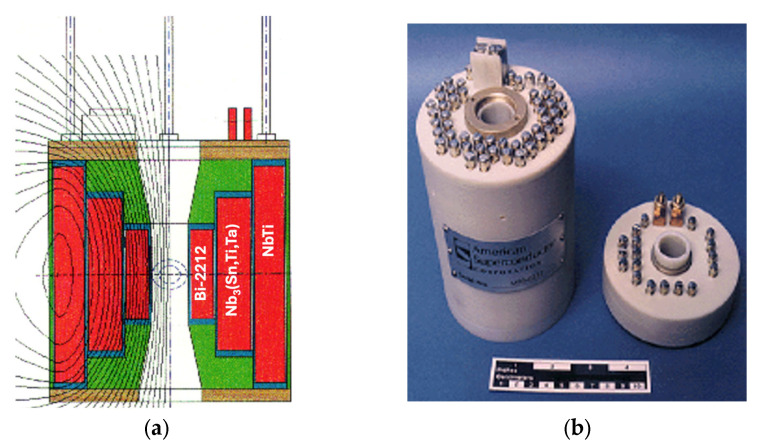
Examples of electromagnets incorporating multiple superconducting materials: (**a**) Composite magnet with an external NbTi winding, a central Nb_3_(Sn,Ti,Ta) winding, and an internal Bi-2212 coil. The Bi-2212 coil contributes an additional 1.46 T to a background field of 22.59 T generated by the combined low-temperature superconductors (LTSs). (**b**) Photograph of American Superconductor high-temperature superconducting (HTS) coils featuring long and short pancake coils. Each pancake coil within the magnet has independent external terminals; however, magnetic field stability and uniformity are affected by differing flux anisotropy orientations, as illustrated schematically in Figure 19a.

**Table 1 materials-18-03665-t001:** The thermophysical properties of helium, hydrogen, and nitrogen cryogenic liquids.

Property	Helium (He)	Hydrogen (H_2_)	Nitrogen (N_2_)
Boiling Point @ 1 atm	4.2 K	20.3 K	77.4 K
Latent Heat of Vaporization	20.9 kJ/mol	445 kJ/kg	199 kJ/kg
	≈5.23 kWh/kg	≈123.6 Wh/kg	≈55.3 Wh/kg
Liquid Density	125 kg/m^3^	71 kg/m^3^	808 kg/m^3^
Typical Market Price	30–50 $/L	4–6 $/L	0.10–0.20 $/L
Energy Equivalent	—	~33.3 kWh/kg (HHV)	—

**Table 2 materials-18-03665-t002:** Comparison of d-LH_2_ cooled superconductors: T_c_, cooling, material dependency, toxicity, production complexity. Only four practical superconductors are recognised as safe by EU.

Superconductor	T_c_ (K)	Cooling Medium	Material Dependency	Toxicity & Safety	Production Complexity
Hg-1223 (HTS)	135	LN_2_ (77 K) LH_2_ (20 K)	Mercury (**Hg**)	Extremely toxic (highly restricted)	Difficult (vapor-phase synthesis)
Tl-1223 (HTS)	127	LN_2_ (77 K) LH_2_ (20 K)	Thallium (**Tl**)	Extremely toxic (highly restricted)	Difficult (vapor-phase synthesis)
Bi-2223 (HTS)	110	LH_2_ (20 K)	(**Pb**)-doped	Moderate toxicity	Complex (multi-phase sintering)
Bi-2212 (HTS)	95	LH_2_ (20 K)	Bismuth (Bi)	**Safe**	Complex (multi-phase sintering)
YBCO (HTS)	92	LN_2_ (64 K) LH_2_ (20 K)	Rare earth (Y, Sm, Nd)	**Safe**	Complex (coated conductor, slow deposition)
SmFeAs(O,F) (Iron Pnictide)	55	LH_2_ (20 K)	Arsenic (**As**) Sm, Fe,	toxicity risk	Difficult (thin films, complex doping)
MgB_2_ (HTS)	39	LH_2_ (14 K)	Boron (strategic)	**Safe**	Simple, but low high-field performance in PIT
Nb_3_Ge (LTS)	23	LH_2_ (14 K)	Nb, Ge (abundant)	**Safe**	Metallic synthesis, simpler than HTS

**Table 3 materials-18-03665-t003:** Comparison of T_c_ and B_c2_ of selected A-15 superconductors.

Compound	T_c_ [K]	B_c2_ [T]
Nb_3_Sn	18.5	24
Nb_3_Al	18.9	29.5
Nb_3_(Al,Ge)	21.0	44
Nb_3_Ge	23.2	38

## Data Availability

Not applicable.

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
