# Peer review of "Hydrogen Cryomagnetic a Common Solution for Metallic and Oxide Superconductors"

_materials, 2025, doi:10.3390/ma18153665_

Round 1
Reviewer 1 Report
Comments and Suggestions for Authors
This is an excellent review article on a very current topic.
Most of the questions, comments, and suggestions can be found in the attached document.
It would be nice if the article were added with another point that is only touched on very superficially.
Namely, it would be important to clarify which superconductor materials can be cooled directly or indirectly, and why.
The "ballooning effect" often plays a role here. Due to their manufacturing method (powder in tube), MgB2 conductors are often enclosed in a stainless steel tube (powder in tube), which can then, of course, be cooled directly with liquid hydrogen. In ceramic-based tape conductors, liquid hydrogen can penetrate and cause this ballooning effect upon warm-up of the system.
It would be nice if criteria were listed here as to when and why superconductors can or must be cooled directly or indirectly.

Author Response
B.A.Glowacki for Referee 1 answers
Answers and clarification of questions projected by to the Referee 1
Q: In referee comment the first suggestion was that it would be important to clarify which superconductor materials can be cooled directly or indirectly, and why.
A: I grateful to Referee for raising this question because it is an important aspect of the cooling strategy for Liquid Hydrogen as a coolant which needs to be explained in detail. As it was discussed in §2.1 line 110-113 : “With growing global investment and infrastructure, LHâ‚‚ is expected to become widely available. This enables the broader adoption of indirect LHâ‚‚ cooling (i-LHâ‚‚) as a standard solution for both metallic and oxide superconductors under the emerging domain of hydrogen cryomagnetic.”
i-LH2 The safety aspect at the initial stage of the Hydrogen Cryomagnetic development
This sentence was referring to examples of i-LH2 presented in Fig. 4 and Fig.5. In Figure 4, the precooled (by LH2) compressed helium gas directly cools to electromagnet via closed He gas installation equipped with cryo-fan. In Figure 5 there is an example of conduction cooling of superconducting coil being in direct contact with inner metal wall of the LH2 cryostat. Both examples of i-LH2 are emphasising safety and can be used at the initial stage of the Hydrogen Cryomagnetic development. The i-LH2 approach will the best option.
d-LH2 direct cooling d-LH2 raise concerns of the Referee “In ceramic-based tape conductors, liquid hydrogen can penetrate and cause this ballooning effect upon warm-up of the system”.
In nineties in my lab, we have been conducting a lot of testing of Bi2212 and Bi2223 as well as TlHg-1223 conductors in silver and silver alloy matrix in Liquid nitrogen under temperature regime of 64K-124K for the cable and transformer application. [B.A. Glowacki, E.A. Robinson, S.P. Ashworth An apparatus for the transport current versus magnetic field measurements in LN2 in the temperature range 77–124 K, Cryogenics Volume 37, Issue 3, March 1997, Pages 173-175]. We observed ballooning effect upon warm-up in mentioned above conductors because of penetration of the LN2 under pressure through the micro-perforations in the silver matrix into the porous superconductive core. The mobility of the silver atoms at the silver–ceramic interface promotes the mobility of ceramic atoms, apparently reducing the binding energy of the ceramic inducing required texturing. In addition, the selective adsorption at the solid–liquid interface reduces the surface energy [B.A.Glowacki Texture development of HTS powder-in-tube conductors Supercond. Sci. Technol. 11 (1998) 989–994.]. For example, surface diffusion coefficients as large as 1 cm2 s−1 have been measured on gold and silver in the presence of lead, thallium and bismuth vapour [Dobrzynski L 1978 Handbook of Surfaces and Interfaces (Lille: Levy) p 199] . The high angle grain boundaries of polycrystalline silver and grain growth combined with intergranular defects resulted in micro perforations of the metal matrix. The micro-perforations along the high angle boundaries enabled penetration of the compress LN2 into the volume of superconductive material and subsequently by restricted access and volume expansion of LN2 once heated up cause ballooning effect upon warm-up.
At that time, we have not reported the negative effect of ballooning effect to allow companies manufacturing such conductors to provide solution to seal the micro-perforations of silver or Au-Ag alloy. To solve such a problem of micro perforations is rather simple by soldering it and cover by electric insulation which makes such conductors impregnable to hydrogen.
In case of (RE)BCO coated conductors the additional electric stabilisation and insulation do protect dense superconductors from contact with LH2
At 20 K, the diffusion of atomic hydrogen is effectively frozen, so embrittlement is not an active degradation mechanism during operation in cryogenic LHâ‚‚ environments. Also At 20 K, these values of H2 diffusivity through Polmer insulation drop by at least 6–10 orders of magnitude, rendering hydrogen diffusion insignificant.
(Kinetic (Collision) Diameter is the most relevant value when considering diffusion, permeability, and flow through micro perforations. Collision diameter of Nâ‚‚: 0.364 nanometers (nm) for H2 molecule ~0.289 nm).
In line 196 the following sentence was added (without going to the details of ballooning effect).
In conclusion all currently manufactured commercially available superconductors can be cooled indirectly i-LH2 and directly d-LH2. This was a reason that in Table 1 it is specifically pointed out that all Superconductive materials can be d-LH2 cooled. It is the safety that define which cooling method will be used for a given application.
In the body of the PDF text Referee placed valuable comments which I clarify below
Page 1 line 32
Q: cost efficiency of LH2 in comparison with LHe cooling versus safety.
A: As I have indicated in the text there is growing global investment and infrastructure, and LHâ‚‚ is expected to become widely available.
There are some reasons for low-cost hydrogen
Recent development of ‘membrane free’ electrolysis makes near future production cost via water electrolysis 7 times cheaper than conventional electrolysis.
Cost of 1 l of liquid hydrogen ~ 30p where cost of 1 litre of LHe 45 £. Latent heat of vaporisation of 1 l of LHe 20.9 kJ/kg; 1l of LH2 446 kJ/kg. LH2 as an energy storage and coolant is a viable option. And there is no future for LHe with superconductivity considering Fig 1, Fig 3 and Fig 4 where cooling by LHe is 2000 times more expensive than by LH2.
It important to remember that every LHe dewar is suitable for storying LH2 which is an order of magnitude cheaper and safer that high pressure installation for H2 gas.
Storage of liquid hydrogen in a car and ortho/para conversion has been solved 20 years ago by BMW /Linde and even 300atm LH2 dewars were build and installed in car. One can be also sure that future cars will be electric but powered by hydrogen and Fuel cells, not batteries.
There is safety related to use and storage of hydrogen, but the general approach “let it go” as a smallest molecule is the common safety approach in industry.
In scenarios of installations where excess of solar and wind energy generated and is not utilised, cost of hydrogen production and liquefaction has no real economic reference and can be treated as “free” or “not lost”.
Page 4 line 122
A: I am grateful to Referee for pointing out that in figure caption of Fig 3 it now hydrogen instead of helium.
Page 8 line line 254
Q: What is energy value of LN2
A: The sentence in line 253 : However, while LNâ‚‚ is widely available and low-cost, it has no energy value, and many HTS conductors suffer significant reductions in J–B–T performance at its boiling point
Is replaced with sentence :
However, while LNâ‚‚ is widely available and low-cost, it has no intrinsic energy value as a fuel or chemical energy carrier it cannot release usable energy through combustion or reaction, unlike hydrogen, which is a true energy carrier capable of releasing significant energy when oxidized. (The only recoverable energy from LNâ‚‚, analogous to that of LHâ‚‚, comes from its physical expansion during the phase change from liquid to gas. This expansion can be harnessed via a turbine to generate electricity on demand). Moreover, many high-temperature superconducting (HTS) conductors experience significant degradation in their J–B–T performance envelope at the boiling point of LNâ‚‚.
Page 10 line 345 fig 9
Q: In Fig. 9 a, the magnetic field is denoted by H and in Fig. 9 b by B(T). A consistent nomenclature throughout the article would be desirable.
A: I would like to mention that I am grateful to referee for this comment. In superconductivity society very often people use Magnetic Field in Tesla which is incorrect and it should be in A/m . Strictly speaking it should be Magnetic Flux density (T) or Magnetic Induction (T) I tried to reform my colleagues and brings some common sense by introducing Magnetic Field as moH in units (T) so it is principally correct.
The origin of the current problem with units for superconductivity originated from the Meissner state where it was CGS system before SI where Gauss = Oersted. So scientist start to use interchangeable Bc1 , Hc1.
I agree that in Fig 9a and the corresponding text I have changed from H to B
I decided that removing Figure 9a and shortening the figure caption will be more beneficial for the future reader.
Page 16 Line 558-560
Q: From today's perspective, it is questionable to what extent an MRI cooled with liquid hydrogen can be operated in a hospital. This aspect should also be considered.
A: A Typical MRI system operated with GM / pulse tube cryocooler as a “zero boil off” or direct conduction in mobile unit or stationary unit. During MRI scan cryocooler is switch off to reduce possible noise. In Hydrogen cryomagnetic society LH2 will be serving as a coolant, energy storage and energy source and can benefit hospital from peripheral aspects of cryogenics. Every MRI installed in building has a duct for emergency evacuation of helium gas in case of magnet quench or failure. Therefore, such duct is going to be used to provide line of compress He gas with precooling from i-LH2 heat exchange with an additional cooling by JT (For instance, helium-based cryocoolers can supplement i-LHâ‚‚ systems to achieve operating temperatures in the 4 K–14 K range for existing NbTi-based MRI).
I would have replaced LH2 with i-LH2 in line 560 to bring clarity to the sentence so potential reader will not think that we are suggesting to fill MRI magnets with LH2. (Not yet).
Also I will add new explanatory text : A typical MRI system uses a GM or pulse-tube cryocooler to operate in “zero boil-off” mode or via direct conduction, particularly in mobile or stationary installations. During scanning, the cryocooler is often turned off to minimize acoustic and electromagnetic noise. In a hydrogen-based cryomagnetic infrastructure, liquid hydrogen (LHâ‚‚) can serve simultaneously as a coolant, energy store, and energy source, offering additional hospital-wide benefits through cryogenic integration. Existing MRI facilities already include helium exhaust ducts for quench events; these can be repurposed to accommodate compressed helium close circuit lines, with precooling via LHâ‚‚ heat exchange, and optionally, final cooling using a Joule–Thomson (JT) stage. Such configurations enable efficient operation in the 4–14 K range for NbTi-based MRI systems, enhancing the sustainability and resilience of clinical cryogenics.
Page 19 , Figure 18
Q: In this figure, the LH2 range extends from 14 K to 21 K. However, since the critical point of hydrogen is 32.9 K and 12.84 bar (Paara-Hydrogen), the range would have to be extended to 32 K. This applies analogously to helium.
A: Of course I agree with referee , but my concern is that however raising of the temperature of LH2 and LHe by presurrising it is a physical fact , but article is about LTS as well and it is important to focus reader mind on the 20K and below where performance of superconducting materials can be at it best.
To compromise I implemented alteration to the Figure 18 where two ranges of the temperatures for existence of LH2 are marked as under lower pressure and under higher pressure. Also, for the LHe existence range was expanded.
Page 21 line 756
Q: What does it mean that liquid nitrogen isn't an energy carrier? Thermodynamically, that's not correct.
The performance at 20 K, enabled by liquid hydrogen (LH2) as an energy carrier and cryogen, is clearly superior to that at 77 K with liquid nitrogen (LN2), which is a non-energy carrier cryogen
A: Because it was already clarified in point on page 8 I suggest removing the part of the sentence shortening to :
The performance at 20 K, enabled by liquid hydrogen (LH2) as an energy carrier and cryogen, is clearly superior to that at 77 K with liquid nitrogen (LN2).
Please see the attachment.

Reviewer 2 Report
Comments and Suggestions for Authors
Interesting work, written in good English which focuses on compatibility of different superconducting materials with hydrogen-based cryogenic system. My questions and remarks are the following:
- First paragraph of Introduction (lines 21-28) – author should give references when mentioning about developments in superconductivity.
- Figure 1 – author shows Monte Carlo simulation from 1990 about helium market price. How about today’s helium market price? Did this simulation predict this right? Please give some comment.
- Figure 2 – author shows power vs. temp. plot. Is this author’s own research? If not, please give the reference.
- Figure 7 – I found the term “superconducting structures” confounding (line 248). That is not the structure superconducts but the compound which hold a specific structure. Besides the YBCO-structure is also common in layered cobaltites which are not superconducting, so we cannot say about specific structure “superconducting”. Also author should give references to this structures.
- Table 1 – in my opinion author should give references to the values of Tc’s presented.
- Figure 9b – author should add references to the presented J-B-T diagrams of MgB2, Nb3Sn, Nb-Ti.
- Figures 8, 15, 18 – I am not sure it is author’s own research. Please confirm or give the reference to this figures
Author Response
B.A.Glowacki for Referee 2 answers
Q1: First paragraph of Introduction (lines 21-28) - author should give references when mentioning about developments in superconductivity.
A1: Referee is absolutely right that I should provide references to support the following sentence
The development of metallic low temperature superconductors, (LTS), and oxide high temperature superconductors, (HTS), has facilitated the realisation of several practical applications, including traction transformers [ ], power cables [ ], fault current limiters, (FCLs) [ ], magnetic ingot heaters [ ], magnetic separators [ ], superconducting energy storage devices [ ], (SMES), motors [ ] , and Tokamak fusion reactors [ ] and many others.
Q2: Figure 1 - author shows Monte Carlo simulation from 1990 about helium market price. How about today's helium market price? Did this simulation predict this right? Please give some comment.
A2: New text was added to clarify the discrepancy after 2 decades . :
‘System dynamics and Monte Carlo simulations were successfully employed to predict helium price trends, showing excellent agreement with historical market data over nearly two decades. This research, based on datasets from BOC and the Culham Fusion Centre, demonstrated that until 2009, the simulated helium prices closely tracked real-world values. However, after 2009, actual helium prices began to diverge sharply from the predicted trajectory, rising far more steeply than anticipated. As of today, helium prices have escalated to such an extent that simulation-based forecasting is no longer meaningful, current prices are so disproportionately high that even helium extraction from ambient air is being reconsidered as a viable option. This deviation is driven by multiple factors: the rise of shale gas production, which contains negligible helium and thus excludes helium recovery from consideration, and the overwhelming global demand for natural gas. As a result, helium recovery during LNG production has become a secondary priority, further tightening global helium supply.’
Q3: Figure 2 - author shows power vs. temp. plot. Is this author's own research? If not, please give the reference.
A3: It is my own research finding of the minimum power requirements for cooling and equipment
Q4: Figure 7 - I found the term "superconducting structures" confounding (line 248). That is not the structure superconducts but the compound which hold a specific structure. Besides the YBCO-structure is also common in layered cobaltites which are not superconducting, so we cannot say about specific structure "superconducting". Also author should give references to this structures.
A4: I am grateful to Referee for such observation. In a figure caption : “Figure 7. Images of applied superconducting structures showcasing metallic superconductors NbTi, …..” I should be more precise and state that:
Figure 7. Images of crystallographic structures showcasing metallic superconductors NbTi (bcc crystal structure), Nb₃Sn (A15 crystal structure), and MgBâ‚‚ (hexagonal AlB2-type crystal structure) alongside oxide ceramic superconductors YBaâ‚‚Cu₃O₇₋ₓ (layered perovskite- like structure) and (Bi,Pb)â‚‚Srâ‚‚Caâ‚‚Cu₃Oâ‚“ (Bi-2223) (layered perovskite-like structure with ‘superconducting’ blocks and ‘charge reservoir’ block) [Shamray V.F, Mikhailova A.B., Mitin A.V. Crystal structure and superconductivity of Bi-2223 ISSN 1063-7745, Crystallography Reports, 2009, Vol. 54, No. 4, pp. 584–590. © Pleiades Publishing, Inc., 2009. Original Russian Text © V.F. Shamray, A.B. Mikhailova, A.V. Mitin, 2009, published in Kristallografiya, 2009, Vol. 54, No. 4, pp. 623–629]. Individual superconducting compound Tc value as well as structure induced superconductive anisotropy, γ value are presented. (Images reproduced with permission of IRC in Superconductivity in Cambridge).
Q5:Table 1 - in my opinion author should give references to the values of Tc's presented.
A5: In table 1 references to established Tc values which is widely accepted for representative superconductors are not essential in this particular table. What is the most important that in all cases LH2 can be cooling and toxicity restrict use of the best superconductors narrowing list of applied superconductors to family of four. Also, important general production complexity comparison favours metallic superconductors.
Q6: Figure 9b - author should add references to the presented J-B-T diagrams of MgB2, Nb3Sn, Nb-Ti.
A6: For clarity in this case reference was added as a source of the images [Wozniak M,
High engineering critical current density MgB2 wires and joints for MRI applications, PhD degree at Department of Materials Science and Metallurgy, University of Cambridge 2012].
Q7: Figures 8, 15, 18 - I am not sure it is author's own research. Please confirm or give the reference to this figures.
A7: I am grateful to Referee for his suggestion. All figures in Fig 8 and also Figure 18 are the ownership of the author
Figure 15 has 2 curves one based on JC data provided by [Bertlett et al ] and and the Jc/KW was calculated by author using his elaborated data presented in Fig. 2. Therefore, Figure caption of Fig 15 is modified as follows:
Figure 15 Dependence of the critical current density normalized by cooling power (Jc/kW) as a function of temperature in the range 13 K to 21 K, alongside the corresponding critical current density (Jc) versus temperature in the range 14 K to 21 K, based on data from Bartlett, Laquer, and Taylor [IEEE Trans. Magn., vol. MAG-11, no. 2, 1975, pp. 405–407]. The figure follows the formatting conventions of Figure 8(a) and specifically illustrates performance characteristics for Nb₃Ge. The analysis reveals an optimal operational region around 15 K, where superconducting performance (Jc) remains high while the cooling power requirement remains economically favourable. The shaded blue area indicates the temperature range where hydrogen exists in its liquid phase under atmospheric pressure, between its melting and boiling points, emphasizing the practical relevance of liquid hydrogen cooling for intermediate-temperature superconductors such as Nb₃Ge.
Please see the attachment.

Reviewer 3 Report
Comments and Suggestions for Authors
- In the introduction, it would be good to compare the thermophysical properties of LH2 and LHe liquids. To discuss the advantages of the higher heat capacity of LH2 compared to LHe.
- The author considers obtaining helium only from the air. And it does not mention the main source of obtaining gaseous helium from the accompanying gas during extraction and separation into natural gas components.
- Figure 1 In the caption, the notation is mixed up. The red dotted line with dots is called dotted, and the dotted black line is called dotted.
- Schedule 2. It is necessary to specify the range of operating temperatures for cryoyidrargirum equipment. It would be nice to put the same curves for hydrogen cooling on the graph.
- Give an approximate cost of cryo-hydrogen equipment and cooling capacities.
- For Figures 8, 12, and 14, give the temperatures for which they are constructed.
Author Response
B.A.Glowacki for Referee 3 answers
Q1: In the introduction, it would be good to compare the thermophysical properties of LH2 and LHe liquids. To discuss the advantages of the higher heat capacity of LH2 compared to LHe.
A1: I am grateful to Referee for his comments and to satisfy clarity of presentation the following new table and text were added to introduction section in line 29 :
The thermophysical properties of liquid hydrogen listed in Table 1, particularly its low boiling point, high specific energy, and substantial latent heat, make it a uniquely capable medium among other cryogenic liquids such as helium and nitrogen for combined energy storage and cryogenic applications. When leveraged in superconducting systems, LHâ‚‚ can reduce cooling complexity, lower energy consumption in multistage refrigeration, and enable the integration of fuel and thermal management into a single infrastructure. As decarbonized energy systems advance, these multifunctional properties position LHâ‚‚ as both a key energy carrier and an enabling refrigerant for emerging superconducting technologies.
Property |
Helium (He) |
Hydrogen (Hâ‚‚) |
Nitrogen (Nâ‚‚) |
Boiling Point @ 1 atm |
4.2 K |
20.3 K |
77.4 K |
Latent Heat of Vaporization |
20.9 kJ/mol |
445 kJ/kg |
199 kJ/kg |
|
≈ 5.23 kWh/kg |
≈ 123.6 Wh/kg |
≈ 55.3 Wh/kg |
Liquid Density |
125 kg/m³ |
71 kg/m³ |
808 kg/m³ |
Typical Market Price |
30–50 $/l |
4–6 $/l |
0.10–0.20 $ /l |
Energy Equivalent |
— |
~33.3 kWh/kg (HHV) |
— |
Q2: The author considers obtaining helium only from the air. And it does not mention the main source of obtaining gaseous helium from the accompanying gas during extraction and separation into natural gas components.
A2: Of course we know that he is in the natural gas, but As of today, helium prices have escalated to such an extent that simulation-based forecasting is no longer meaningful, current prices are so disproportionately high that even helium extraction from ambient air is being reconsidered as a viable option. This deviation is driven by multiple factors: the rise of shale gas production, which contains negligible helium and thus excludes helium recovery from consideration, and the overwhelming global demand for natural gas. As a result, helium recovery during LNG production has become a secondary priority, further tightening global helium supply. The situation is so difficult that extraction of helium from the air appears also as an option. Of course as I was discussing in the body of the text technology of extraction from air maybe expensive but if the price of he will raise further than this may trigger the investment.
Q3: Figure 1 In the caption, the notation is mixed up. The red dotted line with dots is called dotted, and the dotted black line is called dotted.
A3: In the figure caption it is stated ‘The red dotted-dashed line represents historical helium price data, while the broken line shows the estimated cost of helium extraction from air’. , which is correct , but for for better clarity the word dashed has been added :
The figure caption section in line 60 reads as follows :
‘The red dotted-dashed line represents historical helium price data, while the broken dashed line shows the estimated cost of helium extraction from air’
Q4: Schedule 2. It is necessary to specify the range of operating temperatures for cryoyidrargirum equipment. It would be nice to put the same curves for hydrogen cooling on the graph.
A4: Referee raised important point which I will address in a separate publication under preparation concerning discussion of data outlined in this manuscript on Fig 2 and Fig 15 for all superconducting materials (listed in ‘old’ table 1 line 271) in the context of hydrogen versus Joule-Thomson (JT) valve, Pulse Tube, 100 Gifford-McMahon Cryocooler (GM), Stirling Cryocooler.
Problem is complex because there are new technologies for more efficient electrolysis and liquefaction process for hydrogen which reduces equipment cost as well as can benefit from hydrogen as an energy carrier (see new table 1).
It important to remember that every LHe dewar is suitable for storying LH2 which is an order of magnitude cheaper and safer that high pressure installation for H2 gas storage.
Also if hydrogen will be produced in rate as predicted in Fig 3, the price and availability of hydrogen, combined with more efficient liquefaction (currently liquefaction use ~30% of caloric value of H2) hydrogen will be the cryogen of choice for most of hydrogen cryomagnetic applications, as well as energy carrier and storage.
Q5: Give an approximate cost of cryo-hydrogen equipment and cooling capacities.
A5: This point is very relevant to Question 4 and as a complex analysis of implementation of new hydrogen cryomagnetic technology requires a separate article, whereas presented article is debating applicability of LTS and HTS at LH2 temperatures for the benefit of application of superconductivity interconnected with liquid hydrogen economy.
Q6: For Figures 8, 12, and 14, give the temperatures for which they are constructed.
A6: Figure 8 a) is generic for all superconductors and represent schematic illustration.
Fig 8 b) and Fig 8 c) are for Nb3Sn at 4.2K
Fig 12 and Fig 14 are for 4.2K. Information on the measurements at 4.2 is implemented to the relevant figure captions.
The most important information is that if (as presented in Fig 11), the maximum of pinning force, especially of metallic superconductive materials will be shifted closer to BC2 than metallic superconductors at LH2 temperatures will be play parallel role to HTS in superconductivity application in everyday life.
Please see the attachment.

Round 2
Reviewer 2 Report
Comments and Suggestions for Authors
Author took into account all my remarks. I accept the present form of manuscript for publication.